# Benign Underfitting of Stochastic Gradient Descent

**Tomer Koren**
Blavatnik School of Computer Science
Tel Aviv University and Google Research
tkoren@tauex.tau.ac.il

**Roi Livni**
Department of Electrical Engineering
Tel Aviv University
rlivni@tauex.tau.ac.il

**Yishay Mansour**
Blavatnik School of Computer Science
Tel Aviv University and Google Research
mansour.yishay@gmail.com

**Uri Sherman**
Blavatnik School of Computer Science
Tel Aviv University
urisherman@mail.tau.ac.il

## Abstract

We study to what extent may stochastic gradient descent (SGD) be understood as a "conventional" learning rule that achieves generalization performance by obtaining a good fit to training data. We consider the fundamental stochastic convex optimization framework, where (one pass, *without*-replacement) SGD is classically known to minimize the population risk at rate $O(1/\sqrt{n})$, and prove that, surprisingly, there exist problem instances where the SGD solution exhibits both empirical risk and generalization gap of $\Omega(1)$. Consequently, it turns out that SGD is not algorithmically stable in *any* sense, and its generalization ability cannot be explained by uniform convergence or any other currently known generalization bound technique for that matter (other than that of its classical analysis). We then continue to analyze the closely related *with*-replacement SGD, for which we show that an analogous phenomenon does not occur and prove that its population risk does in fact converge at the optimal rate. Finally, we interpret our main results in the context of without-replacement SGD for finite-sum convex optimization problems, and derive upper and lower bounds for the multi-epoch regime that significantly improve upon previously known results.

## 1 Introduction

Conventional wisdom in statistical learning revolves around what is traditionally known as the bias-variance dilemma; the classical theory stipulates the quality of fit to the training data be in a trade-off with model complexity, aiming for a sweet spot where training error is small but yet representative of performance on independent test data.

This perspective is reflected in the vast majority of generalization bound techniques offered by contemporary learning theory. Uniform convergence approaches [36, 4] seek capacity control over the model function class, and employ uniform laws of large numbers to argue convergence of sample averages to their respective expectations. Algorithmic stability [9, 32] on the other hand, builds on controlling sensitivity of the learning algorithm to small changes in its input, and provides algorithm dependent bounds. Nevertheless, despite the conceptual and technical differences between these two methods, both ultimately produce risk bounds by controlling the training error, and the *generalization gap*. The same is true for many other techniques, including sample compression [17, 2], PAC-Bayes [18, 12], and information theoretic generalization bounds [29, 37, 24], to name a few.

In recent years it has become clear there are other, substantially different, ways to manage the fit vs. complexity trade-off, that are in a sense incompatible with traditional generalization bound techniques. Evidently, heavily over-parameterized deep neural networks may be trained to perfectly

fit training data and generalize well nonetheless [38, 25, 26], thus seemingly disobeying conventional statistical wisdom. This phenomenon has garnered significant attention, with a flurry of research works dedicated to developing new techniques that would be able to explain strong generalization performance of algorithms in this so called interpolation regime (see 6, 8 and references therein). Notably, while these algorithms do not strike a balance between model complexity and fit to the data in the traditional sense, fundamentally, they still minimize the empirical risk as a proxy to test performance.

To summarize, in the classical and modern regimes alike, learning methods are thought of as minimizing some combination of the training error and generalization gap, with reasoning that relies in one way or another on the following trivial, yet arguably most profound, bound:

$$\text{test-error} \leq \text{train-error} + |\text{generalization gap}| . \tag{1}$$

In this work, we focus on stochastic gradient descent (SGD)—the canonical algorithm for training machine learning models nowadays—and ask whether its generalization performance can be understood through a similar lens. We consider the fundamental stochastic convex optimization (SCO) framework, in which it is well known that SGD minimizes the population risk at a rate of $O(1/\sqrt{n})$ [23]. Remarkably, the classical analysis targets the population risk directly, and in contrast with other generalization arguments, at least seemingly *does not* rely on the above bound. This highlights an intriguing question: Are these quantities, so fundamental to learning theory, relevant to the way that SGD "works"? Put differently, is it possible to provide a more "conventional" analysis of SGD that conforms with (1)?

Our main result shows that, perhaps surprisingly, there exist convex learning problems where the above bound becomes vacuous for SGD: namely, SGD minimizes the population risk, but at the same time, it *does not* minimize the empirical risk and thus exhibits constant generalization gap. This accords neither with the traditional viewpoint nor with that of interpolation, as both recognize the empirical risk as the principal minimization objective. We refer to this phenomenon as *benign underfitting*: evidently, SGD underfits the training data, but its classical analysis affirms this underfitting to be *benign*, in the sense that test performance is never compromised as a result. Our construction presents a learning problem where the output of SGD with step size $\eta$ over $n$ i.i.d. training examples is $\Omega(\eta\sqrt{n})$ sub-optimal w.r.t. the best fit possible, and consequently has a generalization gap of the same order. Notably, with the standard step size choice of $1/\sqrt{n}$ necessary to ensure the population risk converges at the optimal rate this lower bound amounts to a constant.

Many previously plausible explanations for generalization properties of this algorithm are thereby rendered inadequate, at least in the elementary convex setup we consider here. First, it is clear that SGD cannot be framed as any reasonable regularized empirical risk minimization procedure for the simple reason that it does not minimize the empirical risk, which challenges the implicit regularization viewpoint to the generalization of SGD. Second, any attempt to explain generalization of SGD by uniform convergence over any (possibly data-dependent) hypotheses set cannot hold, simply because the sample average associated with the very same training set SGD was trained on is not necessarily close to its respective expectation. Finally, as it turns out, SGD provides for a strikingly natural example of an algorithm that generalizes well but is not stable in *any* sense, as the most general notion of algorithmic stability is entirely equivalent to the generalization gap [32].

We then move on to study the generalization gap and empirical risk guarantees of SGD in a broader context. We study the case of non-convex and strongly convex component functions, and present natural extensions of our basic result. In addition, we analyse the variant of SGD where datapoints are sampled with-replacement from the training set, in which case the train error is of course low but perhaps surprisingly the population risk is well behaved. Finally, we make the natural connection to the study of without-replacement SGD for empirical risk minimization, and derive upper and lower bounds for the multi-epoch regime. These last two points are discussed in further detail in the following.

**With vs without-replacement SGD.** We may view one-pass SGD as processing the data via *without*-replacement sampling from the training set, as randomly reshuffling the examples does not change their unconditional distribution. Thus, it is interesting to consider the generalization gap of the closely related algorithm given by running SGD over examples sampled *with*-replacement from the training set. Considering instability (see the supplementary for a detailed discussion) of SGD for non-smooth losses and the fact that this variant targets the empirical objective, a priori it would

seem this algorithm would overfit the training set and not provide strong population risk guarantees. Surprisingly, our analysis presented in Section 4 reveals this is not the case, and that with a certain iterate averaging scheme the population risk converges at the optimal rate. Consequently, it turns out the generalization gap is well bounded, and therefore that this variant constitutes a natural learning rule that is not stable in any sense but the most general one.

**Without-replacement SGD for empirical risk minimization.** The example featured in our main construction implies a lower bound of $\Omega(n^{-1/4})$ on the convergence rate of a single epoch of without-replacement SGD for finite sum optimization problems. In this setting, we have a set of $n$ convex losses and we wish to minimize their sum by running SGD over random shufflings of the losses. While the smooth case has been studied extensively (e.g., [28, 27, 20, 31]), the non-smooth case has hardly received much attention. In Section 5 we extend our basic construction to a lower bound for the multi-epoch regime, and complement it with nearly matching upper bounds.

**Our techniques.** Fundamentally, we exploit the fact that dimension independent uniform convergence does not hold in SCO [32]. This is a prerequisite to any attempt at separating train and test losses of any hypothesis vector, let alone that produced by SGD. Another essential condition is the instability of SGD for non-smooth losses, as any form of stability would immediately imply a generalization gap upper bound regardless of uniform convergence. Our main lower bound draws inspiration from constructions presented in the works of [7] and [1], both of which rely on instability, the latter also exploiting failure of uniform convergence. However, neither of these contains the main ideas necessary to provoke the optimization dynamics required in our example. A crucial ingredient in our construction consists of encoding into the SGD iterate information about previous training examples. This, combined with careful design of the loss function, gradient oracle and population distribution, allows correlating sub-gradients of independent training examples, and in turn guiding the SGD iterates to *ascend* the empirical risk.

## 1.1 Summary of main contributions

To summarize, the main contributions of the paper are as follows:

- **One-pass SGD in SCO.** In Section 3, we study the basic SCO setup where the component losses are assumed to be individually convex, and present a construction where the expected empirical risk and therefore the generalization gap are both $\Omega(\eta\sqrt{n})$. We also provide extensions of our main construction demonstrating;
    - SCO with non-convex component functions may exhibit cases of benign *overfitting*, where $\mathbb{E}\big[F(\widehat{w}) - \widehat{F}(\widehat{w})\big] = \Omega(\eta^2 n)$.
    - In SCO with $\lambda$-strongly convex losses the worst case generalization gap is $\Omega(1/\lambda\sqrt{n})$ for the standard step size choice.
- **With vs without replacement SGD in SCO.** In Section 4, we prove the variant of SGD where the training examples are processed via sampling *with*-replacement from the *training set* minimizes the population risk at the optimal rate, and thus enjoys a generalization gap upper bound bound of $O(1/\sqrt{n})$.
- **Multi-epoch without-replacement SGD.** In Section 5, we study convergence rates of without-replacement SGD for finite sum convex optimization problems. We prove a lower bound of $\Omega(n^{-1/4}K^{-3/4})$ on the optimization error after $K$ epochs over $n$ convex losses, and complement with upper bounds of $O(n^{-1/4}K^{-1/2})$ and $O(n^{-1/4}K^{-1/4})$ for respectively the multi-shuffle and single-shuffle SGD variants.

## 1.2 Additional related work

**Gradient descent, algorithmic stability and generalization.** Closely related to our work is the study of stability properties of SGD. For smooth losses, [14] provide upper bounds on the generalization gap by appealing to uniform stability, yielding an $O(1/\sqrt{n})$ rate for a single epoch of $n$ convex losses and the standard step size choice. In a later work, [7] prove tight rates for uniform stability of SGD in the setting of *non*-smooth losses, establishing these scale substantially worse; $\Theta(\eta\sqrt{n})$ for step size $\eta$ and $n$ training examples. Our work shows that in fact the worst case rate of the generalization gap completely coincides with the uniform stability rate of SGD.

A number of works prior to ours studied the extent to which SGD can be explained by implicit regularization in SCO. [16] study the setup where losses are smooth but only required to be convex in expectation, and show SGD may successfully learn when regularized ERM does not. Prior to

their work, [11] also rule out a wide range of implicit regularization based explanations of SGD in the basic SCO setup with convex losses. On a more general level, our work is related to the study of stability and generalization in modern learning theory, pioneered by [9, 32]. In particular, the failure of (dimension independent) uniform convergence in SCO was established in [32]. The work of [13] improves the dimension dependence in the construction of [32] from exponential to linear in the number of training examples. Notably, the construction featured in our main result requires the dimension to be exponential in the sample size, however the techniques of [13] do not readily extend to our setting. Thus, the optimal dimension dependence for a generalization gap lower bound is left for future work.

**Without-replacement SGD for empirical risk minimization.** A relatively long line of work studies convergence properties of without-replacement SGD from a pure optimization perspective (e.g., [28, 20, 30, 27, 19, 31]). Nearly all the papers in this line of work adopt the smoothness assumption, with near optimal bounds established by [20]. An exception is the paper of [33] where an $O(1/\sqrt{nK})$ upper bound is obtained for $n$ datapoints and $K$ epochs, albeit only for generalized linear models over a bounded domain — notably, a setting where uniform convergence holds. Prior to this thread of research, [22] prove a convergence rate of $O(n/\sqrt{K})$ for non-smooth loss functions that applies for *any* ordering of the losses. To the best of our knowledge, this is also the state-of-the-art result for without-replacement SGD in the non-smooth setting without further assumptions on the loss functions.

**Benign overfitting vs. benign underfitting.** While both benign underfitting and benign overfitting challenge traditional generalization techniques, that postulate the training error to represent the test error, as we discuss above these two phenomena point to very different regimes of learning. In particular, [34] shows that benign overfitting requires distributional assumptions for the interpolating algorithm to succeed. In contrast, we show that benign underfitting happens for SGD in a setting where it provably learns (namely, SCO), without any distributional assumptions. We also point out that Corollary 1 shows benign overfitting *cannot* happen in the setup we consider, hence the two phenomena seem to rise in different setups.

**Explaining generalization of interpolators.** As already discussed, there is a large recent body of work dedicated to understanding why over-parameterized models trained by SGD to zero training error generalize well [6, 8, and references therein]. In particular, the work of [5] aims at explaining the phenomenon for high dimensional linear models. Some recent papers investigate limitations of certain techniques in explaining generalization of interpolating algorithms: [21] show uniform convergence fails to explain generalization of SGD in a setup where the generalization gap is in fact well bounded, thus in sharp contrast to our work; [3] rule out the possibility of a large class of excess risk bounds to explain generalization of minimum norm interpolants. Unlike our work, they study properties of possible risk bounds when benign overfitting occurs, and thus do not pertain to SGD that never benignly overfits in SCO.

## 2   Preliminaries

We consider stochastic convex optimization (SCO) specified by a population distribution $\mathcal{Z}$ over a datapoint set $Z$, and loss function $f : W \times Z \to \mathbb{R}$ where $W \subset \mathbb{R}^d$ is convex and compact. We denote

$$F(w) := \mathbb{E}_{z \sim \mathcal{Z}} f(w; z), \qquad \text{(population loss)}$$

$$\widehat{F}(w) := \frac{1}{n} \sum_{i=1}^{n} f(w; z_i), \qquad \text{(empirical loss)}$$

where $\{z_1, \ldots, z_n\} \subseteq Z$ stands for the training set, which we regularly denote by $S$. We let $w^\star := \min_{w \in W} F(w)$ denote the population minimizer, and $w_S^\star := \min_{w \in W} \widehat{F}(w)$ denote the empirical risk minimizer (ERM). The diameter of $W$ is defined by $\max_{x, y \in W} \{\|x - y\|\}$ where $\|\cdot\|$ denotes the euclidean norm, and $\mathcal{B}_0^d(1) := \{x \in \mathbb{R}^d \mid \|x\| \le 1\}$ denotes the $L_2$ unit ball in $\mathbb{R}^d$. Given a training set $S = \{z_1, \ldots, z_n\} \sim \mathcal{Z}^n$ and a learning algorithm that outputs a hypothesis $\widehat{w}_S$, we define the generalization gap to be the absolute value of the expected difference between test and train losses;

$$\left| \mathbb{E}_{S \sim \mathcal{Z}^n} \left[ F(\widehat{w}_S) - \widehat{F}(\widehat{w}_S) \right] \right|. \qquad \text{(generalization gap)}$$

Throughout most of the paper, we consider one-pass projected SGD over $S$;

$$\text{initialize at } w_1 \in W;$$
$$\text{for } t = 2, \ldots, n: \quad w_{t+1} \leftarrow \Pi_W\left(w_t - \eta g_t\right), \quad \text{with } g_t \in \partial f(w_t; z_t),$$

where $\partial f(w; z)$ denotes the set of sub-gradients of $f(\cdot; z) \to \mathbb{R}$ at the point $w \in W$, and $\Pi_W \colon \mathbb{R}^d \to W$ the projection operation onto $W$.

## 3    A generalization gap lower bound for SGD

In this section, we establish our main result; that there exist convex learning problems where SGD incurs a large optimization error and therefore also a large generalization gap. When losses are convex these two quantities are closely related since in expectation, the empirical risk minimizer cannot significantly outperform the population minimizer (a claim that will be made rigorous shortly after our main theorem). Our construction builds on losses that are highly non-smooth, leading to SGD taking gradient steps that actually *ascend* the empirical objective.

**Theorem 1.** *Let $n \in \mathbb{N}$, $n \geq 4$, $d \geq 2^{4n \log n}$, and $W = \mathcal{B}_0^{2d}(1)$. Then there exists a distribution over instance set $Z$ and a 4-Lipschitz convex loss function $f \colon W \times Z \to \mathbb{R}$ such that running SGD initialized at $w_1 = 0$, with step size $\eta > 0$ over $S \sim \mathcal{Z}^n$ yields;*

*(i) a large optimization error;* $\mathbb{E}\left[\widehat{F}(\widehat{w}_S) - \widehat{F}(w_S^\star)\right] = \Omega\left(\min\left\{\eta\sqrt{n}, \frac{1}{\eta\sqrt{n}}\right\}\right),$

*(ii) a large generalization gap;* $\mathbb{E}\left[\widehat{F}(\widehat{w}_S) - F(\widehat{w}_S)\right] = \Omega\left(\min\left\{\eta\sqrt{n}, \frac{1}{\eta\sqrt{n}}\right\}\right),$

*where $\widehat{w}_S$ is any suffix average of the iterates. In particular, for $\eta = \Theta(1/\sqrt{n})$, the population risk is $\mathbb{E}\left[F(\widehat{w}_S) - F(w^\star)\right] = O(1/\sqrt{n})$, while the generalization gap and training error are both $\Omega(1)$.*

A detailed proof of Theorem 1 is deferred to the supplementary; in the following we provide an informal overview containing its principal ingredients.

*Proof sketch.* Let $Z := \{0, 1\}^d$, and consider a population distribution $\mathcal{Z}$ such that $z(i) = 1$ with probability $\delta$. We will use a loss function of the form

$$f(w; z) := \|z \odot w\| + \phi(w; z),$$

where $\odot$ denotes element-wise product. The high level idea is that the norm component penalizes $w$'s that correlate with the given sample point $z$, and the $\phi$ function (the details of which are left for the supplementary) is tailored so that it drives the SGD iterates precisely to those areas in the $L_2$ ball where it correlates with the training set $\{z_1, \ldots, z_n\}$. In addition, the choice of parameters is such that the population loss is approximately zero over the entire domain.

Taking $d$ sufficiently large compared to $\delta^{-1}$, we ensure that w.h.p., for every round $t \in [n]$ there exist many coordinates $i \in [d]$ with a prefix of ones; $z_1(i) = \cdots = z_{t-1}(i) = 1$. With $\delta$ chosen sufficiently small compared to $n$, we ensure that as long as $i \in [d]$ is any coordinate chosen independently of $\{z_{t+1}, \ldots, z_n\}$, w.h.p. this coordinate will have a suffix of zeros; $z_{t+1}(i) = \cdots = z_n(i) = 0$.

Our goal is to make SGD take steps $w_{t+1} \approx w_t - \eta e_{i_t}$ (where $e_i$ denotes the $i$'th standard basis vector) where $i_t \in [d]$ is a coordinate with the aforementioned property of having a prefix of ones followed by a suffix of zeros. Note that since these steps are taken *after* the prefix of ones has ended, they will inflict large empirical loss from the norm component, but will not be "corrected" by future steps owed to the suffix of zeros. To achieve this, we design $\phi$ so that it encodes the relevant information into the SGD iterates. Specifically, $\phi$ "flags" (using some extra dimensions) all coordinates $i \in [d]$ where a prefix of ones has been encountered. In addition, using another max component in $\phi$ we have that for all such coordinates $i$, $e_i \in \partial f(w_t; z)$ for any example $z$ (as this component in the loss depends only on the iterate $w_t$). In particular, we get that $e_i \in \partial f(w_t; z_t)$. Then, our gradient oracle just returns a subgradient pointing towards one of these coordinates (for convenience, we use the minimal one) which we denote by $i_t$, and SGD makes the desired step.

Notably, the coordinate $i_t$ chosen by the subgradient oracle is independent of future examples, and therefore will have a suffix of zeros w.h.p. Hence, as mentioned, this ensures no gradient signal after round $t$ will be able to correct the empirical risk ascent on $i_t$. Concluding, we have that for the final

iterate $\widehat{w} := w_{n+1}$, we get $\widehat{w}(i_t) = -\eta$ for all $t \in [n]$, therefore

$$\widehat{F}(\widehat{w}) = \frac{1}{n} \sum_{i=1}^{n} f(\widehat{w}; z_i) \approx \frac{1}{n} \sum_{i=1}^{n} \|z_i \odot \widehat{w}\| \approx \|\widehat{w}\| \approx \sqrt{\eta^2 n} = \eta \sqrt{n}.$$

A similar argument requiring a few more technical steps shows the same is true for any suffix average $\widehat{w}$. Noting that $\widehat{F}(0) = 0$, we get that the optimization error is $\Omega(\eta \sqrt{n})$. The implication for the generalization gap follows immediately with the standard step size choice of $\eta = 1/\sqrt{n}$, owed to SGD's population risk convergence guarantee. For an arbitrary step size, the result follows from a simple computation, and the proof is concluded. $\qquad\square$

The magnitude of the generalization gap featured in Theorem 1 stems from the large optimization error, which results in the empirical risk over-estimating the population risk by a large margin. Evidently, for convex losses the converse is always false; the empirical risk will never significantly under-estimate the population risk (a fact that will turn out false when losses are only required to be convex in expectation — see Section 3.1). Indeed, stability of the regularized ERM solution implies the ERM does not perform significantly better on the training set compared to the population minimizer $w^\star$.

**Lemma 1.** *Let $W \subset \mathbb{R}^d$ with diameter $D$, $\mathcal{Z}$ any distribution over $Z$, and $f : W \times Z \to \mathbb{R}$ convex and $G$-Lipschitz in the first argument. Then $\mathbb{E}\left[\widehat{F}(w^\star) - \widehat{F}(w_S^\star)\right] \leq \frac{4GD}{\sqrt{n}}$.*

*Proof.* Denote the regularized ERM by $\widehat{w}_S^\lambda := \arg\min_{w \in W} \left\{ \frac{1}{n} \sum_{i=1}^{n} f_i(w; z_i) + \frac{\lambda}{2} \|w\|^2 \right\}$. Observe,

$$F(w^\star) \leq \mathbb{E} F(\widehat{w}_S^\lambda) \leq \mathbb{E}\widehat{F}(\widehat{w}_S^\lambda) + \frac{4G^2}{\lambda n} \leq \mathbb{E}\widehat{F}(w_S^\star) + \frac{\lambda}{2}D^2 + \frac{4G^2}{\lambda n},$$

where the second inequality follows from stability of the regularized ERM (see Lemma 13). Choosing $\lambda := 2GD/\sqrt{n}$, we get that

$$\mathbb{E}\left[\widehat{F}(w^\star) - \widehat{F}(w_S^\star)\right] = F(w^\star) - \mathbb{E}\widehat{F}(w_S^\star) \leq \frac{4GD}{\sqrt{n}},$$

as claimed. $\qquad\square$

Since the optimization error is always positive, we see that the upper bound given by Lemma 1 implies an upper bound on the difference between the population and empirical risks.

**Corollary 1.** *For any distribution $\mathcal{Z}$ over $Z$ and Lipschitz loss function $f : W \times Z \to \mathbb{R}$ convex in the first argument, running SGD with step size $\eta := 1/\sqrt{n}$ guarantees $\mathbb{E}\left[F(\widehat{w}_S) - \widehat{F}(\widehat{w}_S)\right] \leq O(1/\sqrt{n})$.*

*Proof.* We have,

$$\mathbb{E}\left[F(\widehat{w}_S) - \widehat{F}(\widehat{w}_S)\right] = \mathbb{E}\left[F(\widehat{w}_S) - F(w^\star)\right] + \mathbb{E}\left[\widehat{F}(w^\star) - \widehat{F}(\widehat{w}_S)\right]$$

The population error term on the RHS is $O(1/\sqrt{n})$ by the classical analysis of SGD. The second term is bounded by Lemma 1;

$$\mathbb{E}\left[\widehat{F}(w^\star) - \widehat{F}(\widehat{w}_S)\right] \leq \mathbb{E}\left[\widehat{F}(w^\star) - \widehat{F}(w_S^\star)\right] \leq 4GD/\sqrt{n},$$

and the result follows. $\qquad\square$

In the subsections that follow we continue to study the generalization gap in the context of common variants to the basic SCO setup.

## 3.1 SCO with non-convex components

When we relax the convexity assumption and only require the losses to be convex in expectation, we can construct a learning problem where SGD exhibits a case of benign *overfitting*. In contrast to Theorem 1, here we actually drive the SGD iterates *towards* an ERM solution, thus achieving a low optimization error and an empirical risk that under-estimates the population risk.

**Theorem 2.** *Let $n \in \mathbb{N}$, $n \geq 4$, $d \geq 2^{4n \log n}$, $W = \mathcal{B}_0^{2d}(1)$, and $\eta \leq 1/\sqrt{n}$. Then there exists a distribution $\mathcal{Z}$ over $Z$ and a 4-Lipschitz loss $f : W \times Z \to \mathbb{R}$ where $\mathbb{E}_{z \sim \mathcal{Z}} f(w; z)$ is convex in $w$, such that for any suffix average $\widehat{w}$ of SGD initialized at $w_1 = 0$, with step size $\eta$;*

$$\mathbb{E}\left[F(\widehat{w}_S) - \widehat{F}(\widehat{w}_S)\right] = \Omega(\eta^2 n).$$

The construction and proof of Theorem 2 given in the supplementary follow a methodology similar to that of Theorem 1. Here however, we exploit non convex losses to form an empirical loss landscape where the ERM solution significantly outperforms the population minimizer $w^\star$ (notably, a feat not possible when losses are individually convex, by Corollary 1). Our loss function is defined by $f(w; z) := \sum_{i=1}^{d} z(i) w(i)^2 + \phi(w; z)$, with each component playing a similar role as before. We work with the distribution $z \sim \{0, 1\}^d$ where $z(i) = 1$ w.p. $\delta$, $z(i) = -1$ w.p. $\delta$, and $z(i) = 0$ w.p. $1 - 2\delta$. The intuition is that coordinates accumulating many $-1$'s offer regions in the $L_2$ ball where the empirical risk is "too good" compared to the population risk. We tailor the extra dimensions and $\phi$ in coordination with the $-1$ values so that the sub-gradients guide the SGD iterates towards these regions, in exactly the same manner the construction of Theorem 1 drives the iterates to high loss regions. We note that while the statement of Theorem 2 is specialized to step size smaller than $1/\sqrt{n}$, it may be extended to any step size using arguments similar to those given in the proof of Theorem 1.

### 3.2 SCO with strongly convex components

Our basic construction extends to the strongly convex case by making only technical modification to Theorem 1. The theorem below concerns the standard step size choice for strongly convex objectives. We provide its proof in the supplementary.

**Theorem 3.** *Let $n \in \mathbb{N}$, $n \geq 10$, $d \geq 2^{4n \log n}$, $W = \mathcal{B}_0^{2d}(1)$, and $\lambda \geq 1/\sqrt{n}$. Then there exists a distribution over instance set $Z$ and a 4-Lipschitz, $\lambda$-strongly convex loss function $f : W \times Z \to \mathbb{R}$*

*(i) the optimization error is large; $\mathbb{E}_{S \sim \mathcal{Z}^n} \left[ \widehat{F}(\widehat{w}_S) - \widehat{F}(w_S^\star) \right] = \Omega \left( \frac{1}{\lambda \sqrt{n}} \right)$,*

*(ii) the generalization gap is large; $\mathbb{E}_{S \sim \mathcal{Z}^n} \left[ \widehat{F}(\widehat{w}_S) - F(\widehat{w}_S) \right] = \Omega \left( \frac{1}{\lambda \sqrt{n}} \right)$,*

*where $\widehat{w}_S$ is any suffix average of SGD initialized at $w_1 = 0$, with step size schedule $\eta_t = 1/\lambda t$. Furthermore, the problem instance where this occurs is precisely the $\lambda$ regularized version of the example featured in Theorem 1.*

We note that an immediate implication of the above theorem is that if we seek a generalization gap upper bound for a weakly convex problem by means of regularization (meaning, by running SGD on a regularized problem), we would have to take $\lambda \geq 1$ to guarantee a gap of $O(1/\sqrt{n})$. To see this, note that the generalization gap (of any hypothesis) of the regularized problem is the same as that of the original. On the other hand, taking $\lambda \geq 1$ will of course be detrimental to the population error guarantee. Hence, one cannot circumvent the generalization gap lower bound by regularization without compromising the population error.

We conclude this section with a note regarding stability rates of SGD in non-smooth SCO. Implicit in Theorem 1, is that average stability of SGD coincides with the tight uniform stability rate of $\Theta(\eta \sqrt{n})$ established by [7]. This is because Theorem 1 provides the $\Omega(\eta \sqrt{n})$ lower bound on the most general stability notion, which is precisely the generalization gap [32]. We refer the reader to the supplementary for a more elaborate discussion.

## 4 SGD with vs without replacement

In this section, we consider a different algorithm in the context of the basic SCO setup; SGD over examples drawn *with*-replacement from the *training set*. This is not to be confused with one-pass SGD discussed in Section 3, which corresponds to without-replacement SGD on the training set, or alternatively with-replacement SGD over the *population* distribution. Given a training set $S = \{z_1, \ldots, z_n\} \sim \mathcal{Z}^n$, we define with-replacement projected SGD initialized at $w_1 \in W$ by

$$w_{t+1} \leftarrow \Pi_W (w_t - \eta \widehat{g}_t), \quad \text{where } \widehat{g}_t \in \partial f(w_t; \widehat{z}_t) \text{ and } \widehat{z}_t \sim \text{Unif}(S).$$

Perhaps surprisingly, this version of SGD does not overfit the training data; our theorem below establishes that with proper iterate averaging, the population risk converges at the optimal rate.

**Theorem 4.** *Let $W \subset \mathbb{R}^d$ with diameter $D$, $\mathcal{Z}$ be any distribution over $Z$, and $f : W \times Z \to \mathbb{R}$ be convex and $G$-Lipschitz in the first argument. Let $S \sim \mathcal{Z}^n$ be a training set of $n \in \mathbb{N}$ datapoints drawn i.i.d. from $\mathcal{Z}$, and consider running SGD over training examples sampled with-replacement, uniformly and independently from S. Then, for step size $\eta = \frac{D}{G\sqrt{n}}$ and $\overline{w} := \frac{2}{n+1} \sum_{t=1}^{n} \frac{n-t+1}{n} w_t$, the following upper bound holds;*

$$\mathbb{E} \left[ F(\overline{w}) - F(w^\star) \right] \leq \frac{10 G D}{\sqrt{n}}.$$

*Proof.* Fix a time-step $t \in [n]$, and observe that if we don't condition on $S$, we may view the random datapoint $\widehat{z}_t$ as a mixture between a fresh i.i.d. sample from the population and a uniformly distributed sample from the previously processed datapoints $\widehat{S}_{t-1} := \{\widehat{z}_1, \ldots, \widehat{z}_{t-1}\}$;

$$\widehat{z}_t \mid \widehat{S}_{t-1} = \begin{cases} z \sim \mathcal{Z} & \text{w.p. } 1 - \frac{t-1}{n}, \\ z \sim \text{Unif}(\widehat{S}_{t-1}) & \text{w.p. } \frac{t-1}{n}. \end{cases}$$

With this in mind, denote $\widehat{f}_t(w) := f(w; \widehat{z}_t)$, fix $\widehat{S}_{t-1}$ and observe:

$$\mathbb{E}_{\widehat{z}_t}\left[\widehat{f}_t(w_t) - \widehat{f}_t(w^\star) \mid \widehat{S}_{t-1}\right] = \left(1 - \frac{t-1}{n}\right)\mathbb{E}_{z \sim \mathcal{Z}}\left[f(w_t; z) - f(w^\star; z)\right]$$

$$+ \frac{t-1}{n}\frac{1}{t-1}\sum_{i=1}^{t-1}\widehat{f}_i(w_t) - \widehat{f}_i(w^\star).$$

Rearranging and taking expectation with respect to $\widehat{S}_{t-1}$ we obtain

$$\left(1 - \frac{t-1}{n}\right)\mathbb{E}\left[f(w_t; z) - f(w^\star; z)\right] = \mathbb{E}\left[\widehat{f}_t(w_t) - \widehat{f}_t(w^\star)\right] + \mathbb{E}\left[\frac{1}{n}\sum_{i=1}^{t-1}\widehat{f}_i(w^\star) - \widehat{f}_i(w_t)\right]$$

$$\leq \mathbb{E}\left[\widehat{f}_t(w_t) - \widehat{f}_t(w^\star)\right] + \frac{4GD\sqrt{t}}{n}, \qquad (2)$$

where the inequality follows from Lemma 1. Now, by a direct computation we have $\sum_{t=1}^{n}\left(1 - \frac{t-1}{n}\right) = \frac{n+1}{2}$, which motivates setting $\overline{w} := \frac{2}{n+1}\sum_{t=1}^{n}\frac{n-t+1}{n}w_t$. By convexity of $F$, Eq. (2), and the standard regret analysis of gradient descent [e.g., 15] we now have

$$\mathbb{E}\left[F(\overline{w}) - F(w^\star)\right] \leq \frac{2}{n+1}\sum_{t=1}^{n}\left(1 - \frac{t-1}{n}\right)\mathbb{E}\left[F(w_t) - F(w^\star)\right]$$

$$\leq \frac{2}{n+1}\sum_{t=1}^{n}\mathbb{E}\left[\widehat{f}_t(w_t) - \widehat{f}_t(w^\star)\right] + \frac{2}{n+1}\sum_{t=1}^{n}\frac{4GD\sqrt{t}}{n}$$

$$\leq \frac{2}{n}\mathbb{E}\left[\sum_{t=1}^{n}\widehat{f}_t(w_t) - \widehat{f}_t(w^\star)\right] + \frac{8GD}{\sqrt{n}}$$

$$\leq \frac{2}{n}\left(\frac{D^2}{2\eta} + \frac{\eta G^2}{2}\right) + \frac{8GD}{\sqrt{n}}$$

$$= \frac{10GD}{\sqrt{n}},$$

where the last inequality follows by our choice of $\eta = \frac{D}{G\sqrt{n}}$. $\qquad \square$

Evidently, the averaging scheme dictated by Theorem 4 does little to hurt the empirical risk convergence guarantee, which follows from the standard analysis with little modifications (for completeness we provide a formal statement and proof in the supplementary). Combined with Lemma 1, this immediately implies a generalization gap upper bound for with-replacement SGD. Notably, this shows with-replacement SGD provides for an example of a (natural) algorithm in the SCO learning setup that is not even stable on-average, but nonetheless has a well bounded generalization gap. We refer the reader to the discussion in the supplementary for more details.

**Corollary 2.** *For any distribution $\mathcal{Z}$ and loss function $f : W \times Z \to \mathbb{R}$ convex and Lipschitz in the first argument, running SGD with step size and averaging as specified in Theorem 4 ensures*

$$\left|\mathbb{E}\left[F(\overline{w}) - \widehat{F}(\overline{w})\right]\right| \leq O(1/\sqrt{n}).$$

*Proof.* We have;

$$\left|\mathbb{E}\left[F(\overline{w}) - \widehat{F}(\overline{w})\right]\right| \leq \left|\mathbb{E}\left[F(\overline{w}) - F(w^\star)\right]\right| + \left|\mathbb{E}\left[\widehat{F}(w^\star) - \widehat{F}(w_S^\star)\right]\right| + \left|\mathbb{E}\left[\widehat{F}(w_S^\star) - \widehat{F}(\overline{w})\right]\right|.$$

The first term is upper bounded by convergence of the population risk provided by Theorem 4, the second by Lemma 1, and the third by the standard analysis of SGD (see the supplementary). $\qquad \square$

# 5  Multi-epoch SGD for empirical risk minimization

In this section, we forgo the existence of a population distribution and discuss convergence properties of without-replacement SGD (wor-SGD) for finite sum optimization problems. A relatively long line of work discussed in the introduction studies this problem in the *smooth* case. The work of [20] noted smoothness is a necessary assumption to obtain rates that are strictly better than the $O(1/\sqrt{nK})$ guaranteed by with-replacement SGD for $n$ losses and $K$ epochs, due to a lower bound that follows from the deterministic case (e.g., [10]). Here we establish that smoothness is in fact necessary to obtain rates that are not *strictly worse* than with-replacement SGD. We consider running multiple passes of wor-SGD to solve the finite sum optimization problem given by the objective

$$F(w) := \frac{1}{n} \sum_{t=1}^{n} f(w; t) \tag{3}$$

where $\{f(w; t)\}_{t=1}^{n}$ is a set of $n$ convex, $G$-Lipschitz losses defined over a convex and compact domain $W \subseteq \mathbb{R}^d$. Throughout this section we let $w^\star := \min_{w \in W} F(w)$ denote the minimizer of the objective Eq. (3). In every epoch $k \in [K]$ we process the losses in the order specified by a permutation $\pi_k : [n] \leftrightarrow [n]$ sampled uniformly at random, either once in the beginning of the algorithm (single-shuffle), or at the onset of every epoch (multi-shuffle). Multi-epoch wor-SGD initialized at $w_1^1 \in W$ is specified by the following equations;

$$w_{t+1}^k \leftarrow \Pi_W(w_t^k - \eta g_t^k), \quad \text{where } g_t^k \in \partial f_t^k(w_t^k)$$
$$w_1^{k+1} := w_{n+1}^k,$$

where we denote $f_t^k(w) := f(w; \pi_k(t))$. A near-immediate implication of Theorem 1 is that there exists a set of convex losses on which a single epoch of wor-SGD cannot converge at a rate faster than $1/n^{1/4}$. Theorem 5 presented below extends our basic construction from Theorem 1 to accommodate multiple epochs. The main challenge here is in devising a mechanism that will allow fresh bad gradient steps to take place on every new epoch.

**Theorem 5.** *Let $n, K \in \mathbb{N}$, $K \geq 4, n \geq 4$, $c := 4/(2^{1/K} - 1)$, $d \geq 2^{6n \log(cnK)}$, and $W = \mathcal{B}_0^{d'}(1)$ where $d' = (nK + 1)d$. Then there exists a set of $n$ convex, 4-Lipschitz losses such that after $K$ epochs of either multi-shuffle or single-shuffle SGD initialized at $w_1^1 = 0$ with step size $\eta \leq 1/\sqrt{2nK}$, it holds that*

$$\mathbb{E}\left[F(\widehat{w}) - F(w^*)\right] = \Omega\left(\min\left\{1, \eta\sqrt{\frac{n}{J}} + \frac{1}{\eta nK} + \eta\right\}\right),$$

*where $\widehat{w}$ is any suffix average of the last $J$ epochs. In particular, we obtain a bound of $\Omega\left(n^{-1/4}K^{-3/4}\right)$ for any suffix average and any choice of $\eta$.*

The proof of Theorem 5 is provided in the supplementary. The construction in the proof takes the idea that the training set can be encoded in the SGD iterate to the extreme. The loss function and gradient oracle are designed in such a way so as to record the training examples in their full form and order into the iterate. We then exploit this encoded information with an "adversarial" gradient oracle that returns the bad sub-gradients on each gradient step in every new epoch.

Next, we complement Theorem 5 with an upper bound that builds on stability arguments similar to those of the smooth case [20]. Importantly though, lack of smoothness means worse stability rates and necessitates extra care in the technical arguments. Below, we prove the multi-shuffle case, and defer the full details for the single-shuffle case to the supplementary.

**Theorem 6.** *Let $S = \{f(w; t)\}_{t=1}^{n}$ be a set of $n$ convex, $G$-Lipschitz losses over a convex and compact domain $W \subseteq \mathbb{R}^d$ of diameter $D$, and consider running $K \geq 1$ epochs of wor-SGD over $S$. Then, we have the following guarantees:*

*(i) For multi-shuffle, with step-size $\eta = D/(Gn^{3/4}K^{1/2})$, we have*

$$\mathbb{E}\left[F(\widehat{w}) - F(w^\star)\right] \leq \frac{3GD}{n^{1/4}K^{1/2}}.$$

*(ii) For single-shuffle, with step-size $\eta = D/(2Gn^{3/4}K^{3/4})$ and assuming $K \geq n$, we have*

$$\mathbb{E}\left[F(\widehat{w}) - F(w^\star)\right] \leq \frac{10GD}{n^{1/4}K^{1/4}}.$$

In both of the above bounds, $\widehat{w} = \frac{1}{nK}\sum_{k\in[K],t\in[n]} w_t^k$, and the expectation is over the random permutations of losses.

*Proof (multi-shuffle case).* Observe;

$$
\begin{aligned}
F(\widehat{w}) - F(w^\star) &\leq \frac{1}{nK}\sum_{k=1}^{K}\sum_{t=1}^{n} F(w_t^k) - F(w^\star) \\
&= \frac{1}{nK}\sum_{k=1}^{K}\sum_{t=1}^{n} F(w_t^k) - f_t^k(w^\star) \\
&= \frac{1}{nK}\sum_{k=1}^{K}\sum_{t=1}^{n} F(w_t^k) - f_t^k(w_t^k) + \frac{1}{nK}\sum_{k=1}^{K}\sum_{t=1}^{n} f_t^k(w_t^k) - f_t^k(w^\star) \\
&\leq \frac{1}{nK}\sum_{k=1}^{K}\sum_{t=1}^{n} F(w_t^k) - f_t^k(w_t^k) + \frac{D^2}{2\eta nK} + \frac{\eta G^2}{2},
\end{aligned}
$$

with the last inequality following from the standard $nK$ round regret bound for gradient descent [see e.g., 15]. To bound the other term, using Lemma 10, we relate the difference between the without-replacement loss distribution and the full batch objective to the uniform stability rate of SGD, which may then be bounded by applying Lemma 11:

$$
\begin{aligned}
\mathbb{E}\left[F(w_t^k) - f_t^k(w_t^k))\right] &= \mathbb{E}_{\pi_1,\ldots,\pi_{k-1}}\mathbb{E}_{\pi_k}\left[F(w_t^k) - f_t^k(w_t^k)\mid w_1^k\right] \\
&\leq \mathbb{E}_{\pi_1,\ldots,\pi_{k-1}}\left[G\epsilon_{\mathrm{stab}}^{\mathrm{SGD}}(t-1)\right] \\
&= G\epsilon_{\mathrm{stab}}^{\mathrm{SGD}}(t-1) \\
&\leq 2\eta G^2\sqrt{t}.
\end{aligned}
$$

Concluding, we have that

$$
\begin{aligned}
\mathbb{E}\left[F(\widehat{w}) - F(w^\star)\right] &\leq \frac{1}{nK}\sum_{k=1}^{K}\sum_{t=1}^{n}\mathbb{E}\left[F(w_t^k) - f_t^k(w_t^k)\right] + \frac{D^2}{2\eta nK} + \frac{\eta G^2}{2} \\
&\leq \frac{2}{nK}\sum_{k=1}^{K}\sum_{t=1}^{n}\eta G^2\sqrt{t} + \frac{D^2}{2\eta nK} + \frac{\eta G^2}{2} \\
&\leq 2\eta G^2\sqrt{n} + \frac{D^2}{2\eta nK} + \frac{\eta G^2}{2} \\
&\leq \frac{3GD}{n^{1/4}K^{1/2}},
\end{aligned}
$$

where the last inequality follows from our choice of $\eta = D/(Gn^{3/4}K^{1/2})$.

$\square$

### Acknowledgements and funding disclosure

This work was supported by the European Research Council (ERC) under the European Union's Horizon 2020 research and innovation program (grant agreement No. 882396), by the Israel Science Foundation (grants number 993/17, 2549/19, 2188/20), by the Len Blavatnik and the Blavatnik Family foundation, by the Yandex Initiative in Machine Learning at Tel Aviv University, by a grant from the Tel Aviv University Center for AI and Data Science (TAD), and by an unrestricted gift from Google. Any opinions, findings, and conclusions or recommendations expressed in this work are those of the author(s) and do not necessarily reflect the views of Google.

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
