# A    Relations to Algorithmic Stability of SGD

In this section, we formally introduce notions of algorithmic stability and relate them to results presented in the paper. Let $Z$ denote a set of datapoints and $\mathcal{Z}$ a distribution over $Z$. For two training sets $S, S' \in Z^n$, we write $S \simeq S'$ if they differ in exactly one datapoint. For a learning algorithm $\mathcal{A} \colon Z^* \to \mathbb{R}^d$, we define the *uniform argument stability* (UAS) of $\mathcal{A}$ by

$$\epsilon_{\text{stab}}^{\mathcal{A}}(n) := \max_{S \simeq S', |S|=n} \left\| \mathcal{A}(S) - \mathcal{A}(S') \right\|, \tag{4}$$

and the *average argument stability* (AAS) of $\mathcal{A}$ by

$$\epsilon_{\text{avgstab}}^{\mathcal{A}}(n) := \max_{\mathcal{Z}} \left\{ \frac{1}{n} \sum_{i=1}^{n} \mathbb{E}_{S \sim \mathcal{Z}^n, z_i' \sim \mathcal{Z}} \left\| \mathcal{A}(S) - \mathcal{A}(S^{(i)}) \right\| \right\}, \tag{5}$$

where $S^{(i)}$ is formed by taking $S$ and replacing $z_i$ with $z_i'$.

It is well known (e.g., [9, 32]) that for any distribution $\mathcal{Z}$ and algorithm $\mathcal{A}$, the following relations holds between the generalization gap, AAS, and UAS;

$$\left| \mathbb{E}_{S \sim \mathcal{Z}^n} \left[ F(\mathcal{A}(S)) - \widehat{F}(\mathcal{A}(S)) \right] \right| \leq \epsilon_{\text{avgstab}}^{\mathcal{A}}(n) \leq \epsilon_{\text{stab}}^{\mathcal{A}}(n). \tag{6}$$

In [7] it was established the UAS of both with and without-replacement SGD is $\Omega(\eta\sqrt{n})$ for $n$ steps of size $\eta$ with $n$ training examples. However, considering Eq. (6), it remained unclear whether the AAS and generalization gap of these algorithms exhibit rates of similar order, in which case the UAS accurately captures the rate of the generalization gap. Interestingly, the answer to this question depends on whether sampling is done with or without replacement, as we discuss next.

**Stability of without-replacement SGD.** As an immediate corollary of our Theorem 1, we have that the AAS of without-replacement SGD is also $\Omega(\eta\sqrt{n})$. This follows from Eq. (6) and since the theorem establishes the generalization gap to be $\Omega(\eta\sqrt{n})$. Similarly, the lower bound given by Theorem 3 demonstrates that the AAS of SGD in the strongly convex case is $\Omega(1/\lambda\sqrt{n})$. Combined with the naive upper bound argument for uniform stability of $O(1/\lambda\sqrt{n})$ (which follows by convergence of SGD iterates to the minimizer in parameter space), we get a tight characterization of stability for strongly convex losses for the standard step size schedule.

**Stability of with-replacement (one-pass) SGD.** In Section 4, specifically in Corollary 2, we establish a generalization gap of $O(1/\sqrt{n})$ for with-replacement SGD with a particular averaging scheme and a properly tuned step size. However, as it turns out, the average argument stability of this version of SGD is nonetheless of order $\Omega(\eta\sqrt{n})$ as we demonstrate in Theorem 7 below. This shows that this version of with-replacement SGD is an algorithm that is not stable in any sense but the most general one (namely, the one being equivalent to the generalization gap).

**Theorem 7.** *Let $n \in \mathbb{N}$, $n \geq 200$, $d \geq 2^{3n}$, $W = \mathcal{B}_0^d(1)$. Further, let $\{\beta_t\}_{t=1}^n$ be an iterate averaging scheme that does not decay too quickly; $\sum_{s=t+1}^{n} \beta_n \geq C((n-t)/n)^2$ for some constant $C > 0$. Then there exists a distribution over the instance set $Z = \{0,1\}^d$ and a 2-Lipschitz, convex loss function $f \colon W \times Z \to \mathbb{R}$ such that for all $k \in [n]$,*

$$\mathbb{E}_{S \sim \mathcal{Z}^n, z_k' \sim \mathcal{Z}} \left\| \widehat{w} - \widehat{w}^{(k)} \right\| \geq \Omega(\eta\sqrt{n}),$$

*where $\widehat{w}, \widehat{w}^{(k)}$ denote the $\{\beta_n\}$-averaged iterates $\{w_t\}, \{w_t'\}$ of $n$ with-replacement SGD iterations (initialized at $w_1' = w_1 = 0$) with step size $\eta > 0$ over the training sets $S$ and $S^{(k)}$ respectively;*

$$\widehat{w} := \sum_{t=1}^{n} \beta_n w_t; \quad \widehat{w}^{(k)} := \sum_{t=1}^{n} \beta_n w_t'.$$

Note that the averaging scheme employed in Theorem 4 decays sufficiently slow so as to satisfy requirements of Theorem 7, hence the stability lower bound follows.

*Proof.* Let $\mathcal{Z}$ be defined by $z(i) \sim \text{Ber}(1/2)$, and set

$$f(w;z) := -\epsilon \sum_{i=1}^{d} \alpha_z(i) w(i) + \max_{i \in [d]} \{w(i)\}.$$

We will take $\epsilon := \beta_n/(16n^3 d)$, and define

$$\alpha_z(i) = \begin{cases} -n & z(i) = 1, \\ 1 & z(i) = 0. \end{cases}$$

In addition, let

$$I(w) := \arg\min_{i \in [d]} \left\{ i \mid w(i) = \max_j\{w(j)\} \right\},$$

set

$$i_t := I(w_t), \quad i'_t := I(w'_t), \tag{7}$$

and define

$$g(w; z) := -\epsilon \alpha_z + e_{I(w)}, \tag{8}$$

where $e_i$ denotes the $i$'th standard basis vector. It follows that $g(w; z) \in \partial f(w; z)$ and $\|g(w; z)\| \leq nd\epsilon + 1 \leq 2$ for all $z \in Z, w \in W$. Proceeding, we denote

$$S = \{z_1, \ldots, z_n\}, \quad S' = \{z'_1, \ldots, z'_n\},$$

and note that $z_l = z'_l$ for all $l \neq k$. Further, we denote the training examples sampled by SGD by

$$\widehat{z}_t := z_{k_t} \in S, \quad \widetilde{z}'_t := z'_{k_t} \in S^{(k)},$$

where $\{k_t\} \sim \text{Unif}[n]$ are uniformly random and independent training indices. For the remainder of the proof, we condition on the event

$$\mathcal{E} = \{(z_1, \ldots, z_n, z'_k) \mid Z(S) \geq n, \text{ and } Z(S') \geq n\}, \tag{9}$$

where $Z(S) := |\{i \mid \forall r \in [n], z_r(i) = 0\}|$ and similarly $Z(S') := |\{i \mid \forall r \in [n], z'_r(i) = 0\}|$. Owed to our assumption that $d \geq 2^{3n}$, a standard concentration argument shows this event occurs with probability $\geq 1/2$.

We will now proceed to track how the SGD iterates evolve. Observe that for all $t \in [n]$, we have by direct computations of the gradient steps with Eq. (8);

$$i \notin \{i_1, \ldots, i_t\} \implies w_{t+1}(i) = \eta\epsilon \sum_{s=1}^t \alpha_{\widehat{z}_s}(i)$$

$$\implies \begin{cases} w_{t+1}(i) = \eta\epsilon t & \forall s \leq t, \ \widehat{z}_s(i) = 0, \\ w_{t+1}(i) \in [-\eta\epsilon n^2, 0] & \exists s \leq t, \ \widehat{z}_s(i) = 1. \end{cases}$$

In addition, from similar computations;

$$i \in \{i_1, \ldots, i_t\} \implies w_{t+1}(i) \leq -\eta + \eta\epsilon n.$$

Summarizing, and applying identical calculations for $w'_{t+1}$, we have:

$$t < s \implies w_s(i_t) \leq -\eta + \eta\epsilon n, \quad i \notin \{i_1, \ldots, i_n\} \implies \forall s, \ w_s(i) \in [-\eta\epsilon n^2, \eta\epsilon n]$$

$$t < s \implies w'_s(i'_t) \leq -\eta + \eta\epsilon n, \quad i' \notin \{i'_1, \ldots, i'_n\} \implies \forall s, \ w'_s(i') \in [-\eta\epsilon n^2, \eta\epsilon n]. \tag{10}$$

By Eq. (10) above, for all $t \in [n]$ we have;

$$\widehat{w}(i_t) = \sum_{s=1}^n \beta_s w_s(i_t) \leq \sum_{s=1}^t \beta_s \eta s\epsilon - \sum_{s=t+1}^n \beta_s(\eta - \eta\epsilon n)$$

$$\leq -\eta \sum_{s=t+1}^n \beta_s + \eta n\epsilon \sum_{s=1}^n \beta_s \leq -\frac{3}{4}\eta \sum_{s=t+1}^n \beta_s, \tag{11}$$

where the last inequality follows from our choice of $\epsilon$. In addition, if $i_t \notin \{i'_1, \ldots i'_n\}$, again by Eq. (10) and our choice of $\epsilon$ it follows that;

$$\widehat{w}^{(k)}(i_t) = \sum_{s=1}^n \beta_s w'_s(i_t) \geq \sum_{s=1}^n \beta_s \eta n^2 \epsilon \geq -\frac{\eta}{4}\beta_n. \tag{12}$$

Now, set $t_0 := \min\{t \mid k_t = k\}$ to be the first time that index $k$ (in which training examples differ) is chosen, and let $t > t_0$. Note that $z'_k(i_t) = 1$ implies $i_t \notin \{i'_1, \ldots, i'_n\}$; to see this, observe that for $\tau \leq t_0, i'_\tau = i_\tau$, while $\tau > t_0$ implies $i'_\tau \neq i_t$, since the event Eq. (9) we condition on ensures

$$i_t = \min\{i \in [d] \mid \forall s < t, \, \widehat{z}_s(i) = 0\},$$
$$i'_t = \min\{i \in [d] \mid \forall s < t, \, \widehat{z}'_s(i) = 0\}.$$

(From the above it also must hold that $i_t \neq i_\tau, i'_t \neq i'_\tau$ for all $t \neq \tau$.) Thus, putting together Eq. (11), Eq. (12) and the fact that $z'_k(i_t) = 1$ implies $i_t \notin \{i'_1, \ldots, i'_n\}$, we obtain for all $t_0 < t$;

$$|\widehat{w}(i_t) - \widehat{w}^{(k)}(i_t)| \geq \mathbb{1}\left\{z'_k(i_t) = 1\right\} \frac{\eta}{2} \sum_{s=t+1}^{n} \beta_s \geq \mathbb{1}\left\{z'_k(i_t) = 1\right\} \frac{\eta C(n-t)^2}{2n^2},$$

where the second inequality follows from our assumption on $\{\beta_t\}$. Thus, for $t_0 < t \leq 3n/4$, we get that

$$|\widehat{w}(i_t) - \widehat{w}^{(k)}(i_t)| \geq \mathbb{1}\left\{z'_k(i_t) = 1\right\} \frac{C}{4}\eta,$$

and taking expectations we obtain;

$$\mathbb{E}\left[\left\|\widehat{w} - \widehat{w}^{(k)}\right\| \mid t_0\right] \geq \mathbb{E}\left[\sqrt{\sum_{t=1}^{n}(\widehat{w}(i_t) - \widehat{w}'(i_t))^2} \mid t_0\right]$$

$$\geq \frac{C\eta}{4}\mathbb{E}\left[\sqrt{\sum_{t=t_0+1}^{3n/4}\mathbb{1}\left\{z'_k(i_t) = 1\right\}} \mid t_0\right].$$

Now, observe that $z'_k$ is independent of $i_t$ for all $t$, hence the expectation above is of the form

$$\mathbb{E}\sqrt{\sum_{l=1}^{m} Y_l},$$

where $m := 3n/4 - t_0$ and $Y_l \sim \mathrm{Ber}(1/2)$ are independent. Thus,

$$\Pr\left(m/2 - \sum_{l=1}^{m} Y_t > m/4\right) \leq e^{-m/16} \leq 1/2,$$

for $m > 100$, and then

$$\mathbb{E}\left[\left\|\widehat{w} - \widehat{w}^{(k)}\right\| \mid t_0\right] \geq \frac{C\eta}{4}\frac{1}{2}\sqrt{3n/4 - t_0} = \frac{C\eta}{8}\sqrt{3n/4 - t_0}.$$

To conclude, we use the fact that $t_0$ follows a geometric distribution with parameter $1/n$, therefore

$$\Pr(t_0 \leq n/2) = \frac{1}{n}\sum_{t=1}^{n/2}(1 - 1/n)^t = 1 - (1 - 1/n)^{n/2} \geq 1 - e^{-1/2} \geq 1/3.$$

This implies,

$$\mathbb{E}\left\|\widehat{w} - \widehat{w}^{(k)}\right\| \geq \frac{C\eta}{32}\sqrt{3n/4 - n/2} = \frac{C\eta}{64}\sqrt{n},$$

and completes the proof. $\square$

## B  Proof of Theorem 1

Our first proof below applies for step sizes $\eta \leq 1/\sqrt{n}$. The extension for larger step sizes is rather technical and requires care of the projection step — we provide it in the supplementary. The statement of Theorem 1 is repeated below for the case of the small step size regime.

**Theorem 8** (Small step size case of Theorem 1)**.** *Let $n \in \mathbb{N}$, $n \geq 4$, $d \geq 2^{4n \log n}$, and $W = \mathcal{B}_0^{2d}(1)$. Then there exists a distribution over instance set $Z$ and a 4-Lipschitz convex loss function $f : W \times Z \to \mathbb{R}$ such that*

(i) *the optimization error is large;* $\mathbb{E}_{S \sim \mathcal{Z}^n} \left[ \widehat{F}(\widehat{w}_S) - \widehat{F}(w_S^\star) \right] = \Omega\left(\eta \sqrt{n}\right)$,

(ii) *the generalization gap is large;* $\mathbb{E}_{S \sim \mathcal{Z}^n} \left[ \widehat{F}(\widehat{w}_S) - F(\widehat{w}_S) \right] = \Omega\left(\eta \sqrt{n}\right)$,

*where $\widehat{w}$ is any suffix average of SGD with step size $\eta \leq 1/\sqrt{n}$.*

*Proof.* Our construction is parameterized by $\epsilon, \delta > 0$, which will be chosen later. We will work with the datapoints set $Z = \{0, 1\}^{2d}$ and define the distribution $\mathcal{Z} = \mathcal{Z}(\delta)$ over $Z$ by

$$\forall i \leq d; \quad z(i) = \begin{cases} 1 & \text{w.p. } \delta; \\ 0 & \text{w.p. } (1 - \delta), \end{cases}$$

$$\forall i > d; \quad z(i) = z(i - d).$$

Our loss function is a combination of two components; the "push" function $\phi$ is in charge of driving the SGD iterate towards areas in the $L_2$ ball where the "penalty" function $v$ inflicts a norm-like loss.

$$\phi(w; z) := -\epsilon \sum_{i=d+1}^{2d} z(i)w(i) + \max_{1 \leq i \leq d} \{w(i) + w(i + d)\}, \tag{13}$$

$$v_z(w) := \sqrt{\sum_{i=1}^{d} z(i)w(i)^2}, \tag{14}$$

$$f(w; z) := \phi(w; z) + v_z(w).$$

The lower bound arguments all go through with any sub-gradient oracle $g(w; z) \in \partial_w f(w; z)$. For clarity of exposition, we make use of the gradient oracle $g$ that returns the minimal coordinate sub-gradient for the max component in $\phi$;

$$g_\phi(w; z)(i) := \begin{cases} \mathbb{1}\{i = I(w)\} & i \leq d \\ -\epsilon z(i) + \mathbb{1}\{i = I(w) + d\} & i \geq d \end{cases}, \tag{15}$$

$$\text{where } I(w) := \min \left\{ i \in [d] \mid i \in \arg\max_{1 \leq j \leq d} \{w(j) + w(j + d)\} \right\}. \tag{16}$$

We additionally denote the index picked by $g$ on round $t \in [n]$ by

$$i_t := I(w_t). \tag{17}$$

We then set $g(w; z) := g_\phi(w; z) + \nabla v_z(w)$. It now follows that for all $w, z \in \mathbb{R}^{2d}$, $g_\phi(w; z) \in \partial_w \phi(w; z)$, thus $g(w; z) \in \partial_w f(w; z)$. Choosing $\epsilon = 1/d$, we get that $f$ is 4-Lipschitz;

$$\|g(w; z)\| \leq \epsilon\sqrt{d} + 2 + \frac{1}{2\|w\|}\|w\| \leq 3 + 1/\sqrt{d} \leq 4.$$

With the above construction in place, we first claim that with sufficiently large probability, the training set will contain the desired collection of "bad" coordinates which will be picked up by our gradient oracle. Indeed, with the dimension $d$ large enough, a proper choice of $\delta$ ensures that for every $t \in [n]$, there will be a certain coordinate with a prefix of $t - 1$ ones followed by a zero only suffix.

*Lemma 2.* For $\delta = 1/4n^2$, with probability $\geq 1/2$ over the random draw of $S = \{z_1, \ldots, z_n\} \sim \mathcal{Z}^n$, it holds that for all $t \in [n]$:

1. *There exist prefix of ones coordinates $E^{(1,t)} := \{j \in [d] \mid s < t \implies z_s(j) = 1\} \neq \emptyset$, and*

2. *the minimal such coordinate $J_t := \min\{j \in E^{(1,t)}\}$ also has a zero suffix; $s \geq t \implies z_s(J_t) = 0$.*

From this point onward fix $\delta := 1/4n^2$. By definition of our gradient oracle, a relatively straightforward argument given in our next lemma establishes SGD will take gradient steps precisely on those bad coordinates of Lemma 2. Notably, we have designed the construction so that these steps are made only after the samples penalizing those coordinates have been processed. This eliminates the possibility for SGD to correct these coordinates in future steps.

**Lemma 3.** *We have with probability $\geq 1/2$ that for all $t \in [n]$, $i_t = J_t$ (see Eq. 17), and for all $\tau \in [n], \tau > t$, $w_\tau(i_t) = -\eta$.*

To complete the proof, we assume the event from the lemma occurs. Since it occurs with constant probability, a lower bound derived conditioned on it implies a lower bound in expectation. First we argue the population loss of all iterates is upper bounded as

$$
\begin{aligned}
F(w_t) &= -\epsilon \sum_{i=d+1}^{2d} \delta w_t(i) + \epsilon n \eta + \mathbb{E}\left[ \sqrt{\sum_{i=1}^{d} z(i) w_t(i)^2} \right] \\
&\leq -\epsilon \delta \sum_{s=1}^{t-1} w_t(i_s) + \epsilon n \eta + \sqrt{\sum_{i=1}^{d} \mathbb{E}[z(i)] w_t(i)^2} \\
&= \epsilon \delta(t-1)\eta + \epsilon n \eta + \sqrt{\delta \sum_{i=1}^{d} w_t(i)^2} \\
&\leq 2\epsilon n \eta + \sqrt{\delta \sum_{s=1}^{t} w_t(i_s)^2} \\
&\leq 2\epsilon n \eta + \sqrt{\delta \eta^2 n} = \frac{2n\eta}{d} + \frac{\eta \sqrt{n}}{2n} \leq \frac{\eta}{\sqrt{n}}.
\end{aligned}
$$

By convexity of the population loss, the above implies that any suffix average satisfies $F(\widehat{w}) \leq \eta/\sqrt{n}$. In addition, note that $\widehat{F}(w_S^\star) \leq \widehat{F}(0) \leq 0$, hence the $\Omega(\eta\sqrt{n})$ bound we will now establish on the empirical risk of SGD implies our claimed optimization and generalization lower bounds. Indeed, let $\tau \in [n]$ and denote $\overline{w}_\tau := \frac{1}{n-\tau+2} \sum_{t=\tau}^{n+1} w_t$. Observe that for $1 \leq t \leq n/2$, by Lemma 3 at least half of the iterates have the $-\eta$ value in coordinate $i_t$;

$$
\begin{aligned}
\overline{w}_\tau(i_t) &= \frac{1}{n-\tau+2} \sum_{s=\tau}^{n+1} w_s(i_t) \leq \frac{1}{n-\tau+2} \sum_{s=\max\{\tau, n/2\}}^{n+1} w_s(i_t) \\
&\leq \frac{n - \max\{\tau, n/2\} + 2}{n-\tau+2} (-\eta) \leq -\frac{\eta}{2}.
\end{aligned}
$$

(We ignore the fact that the last iterate, formally speaking, may have a slightly greater value due to the projection on the last step.) Now, for any $w \in W$,

$$
\begin{aligned}
\widehat{F}(w) &= \frac{1}{n} \sum_{s=1}^{n} f(w; z_s) = \frac{1}{n} \sum_{s=1}^{n} \phi(w; z_s) + \frac{1}{n} \sum_{s=1}^{n} \sqrt{\sum_{t=s+1}^{n} w(i_t)^2} \\
&\geq \frac{1}{n} \sum_{s=1}^{n} \phi(w; z_s) + \frac{1}{5\sqrt{n}} \sum_{t=n/4}^{n} |w(i_t)|,
\end{aligned}
$$

where the second inequality follows from Lemma 15. Noting that $\phi(\overline{w}_\tau; z_s) \geq -\epsilon^2 dn\eta \geq -\eta$ and combining the last two displays we get that

$$
\widehat{F}(\overline{w}_{\tau:n}) \geq \frac{1}{5\sqrt{n}} \sum_{t=n/4}^{n/2} \frac{\eta}{2} - \eta = \frac{\eta(\sqrt{n}-1)}{40},
$$

which completes the proof. $\qquad\square$

*Proof of Lemma 2.* Fix $t \in [n]$, denote

$$
\begin{aligned}
E^{(1,t)} &:= \{i \in [d] \mid z_s(i) = 1 \ \forall s < t\}, \\
E^{(t,0)} &:= \{i \in [d] \mid z_s(i) = 0 \ \forall s \geq t\},
\end{aligned}
$$

and let $J_t \in E^{(1,t)}$ be the minimal element if it is not empty. Note that

$$
\Pr(E^{(1,t)} = \emptyset) = \Pr\left(\forall i \in [d], \ \exists s < t, \ z_s(i) = 0\right) = \left(1 - \delta^{t-1}\right)^d \leq (1 - \delta^n)^d.
$$

In addition, since the contents of $E^{(1,t)}$ are independent of $z_t, \ldots, z_n$, we have that for any $i \in E^{(1,t)}$,

$$\Pr(i \in E^{(t,0)}) = (1 - \delta)^{n-t+1} \geq (1 - \delta)^n.$$

Therefore,

$$\Pr\left(E^{(1,t)} = \emptyset \text{ OR } \left(E^{(1,t)} \neq \emptyset \text{ but } J_t \notin E^{(t,0)}\right)\right) \leq (1 - \delta^n)^d + 1 - (1 - \delta)^n.$$

Now, by the union bound over all values of $t \in [n]$ we obtain

$$\Pr\left(\forall t \in [n], \, E^{(1,t)} \neq \emptyset \text{ AND } J_t \in E^{(t,0)}\right) \geq 1 - n\left((1 - \delta^n)^d + 1 - (1 - \delta)^n\right). \tag{18}$$

Now, since $\delta = 1/4n^2$ we have

$$(1 - \delta^n)^d = \left(1 - \frac{d\delta^n}{d}\right)^d \leq e^{-d\delta^n} \leq \frac{1}{4n},$$

where the last inequality follows for $d \geq \delta^{-n} \log(4n) = 4^n n^{2n} \log(4n)$ (recall that by the assumption in the theorem statement $d \geq 2^{4n \log n} \geq 4^n n^{2n} 4n$). In addition;

$$(1 - \delta)^n = \left(1 - \frac{1}{4n^2}\right)^n \geq 1 - \frac{1}{4n} \implies 1 - (1 - \delta)^n \leq \frac{1}{4n}.$$

Back to Eq. (18), applying the inequalities from the last two displays we obtain the desired event occurs with probability

$$\geq 1 - n\left((1 - \delta^n)^d + 1 - (1 - \delta)^n\right) \geq 1 - n\left(\frac{1}{4n} + \frac{1}{4n}\right) = \frac{1}{2}, \tag{19}$$

and the result follows. $\qquad \square$

*Proof of Lemma 3.* Following a direct computation, we get that

$$g(w; z)(i) = \begin{cases} \mathbb{1}\{i = I(w)\} + \frac{z(i)w(i)}{v_z(w)} & i \leq d, \\ \mathbb{1}\{i = I(w) + d\} - \epsilon z(i) & i > d. \end{cases} \tag{20}$$

From the above we see that the value of $w(i + d)$ for every coordinate $i + d \in \{d + 1, \ldots, 2d\}$ gains $\eta\epsilon$ when $z(i) = 1$, while the value of coordinate $i$ only decreases. Thus $I(w_t)$ will be a coordinate with an all ones prefix if one exists. Formally, let $t \in [n]$, and observe that our gradient oracle will return the minimal coordinate $i_t \in [d]$ with the maximum value of $w_t(i_t) + w_t(i_t + d)$. Assuming the event from Lemma 2 occurs, note that the coordinate $J_t \in [d]$ with $z_1(J_t) = \ldots = z_{t-1}(J_t) = 1$ exists. Now, observe that any coordinate $j \in [d]$ is bound to satisfy

$$w_t(j) + w_t(j + d) \leq w_t(J_t) + w_t(J_t + d).$$

To see this, note that $w_t(J_t) = 0$, because $J_t \neq i_s$ for all $s < t$ (formally this follows by induction). In addition, by Eq. (20);

$$\forall s < t, \, z_s(J_t) = 1 \implies \forall s < t, \, -\eta g(w; z_s)(J_t + d) = \epsilon\eta \implies w_t(J_t + d) = (t - 1)\epsilon\eta.$$

On the other hand, for any $j' \in [d]$ we have $w_t(j') \leq 0$, and $w_t(j' + d) \leq (t - 1)\epsilon\eta$. Concluding, it follows the gradient oracle will pick $i_t = I(w_t) = J_t$, therefore $w'_{t+1}(i_t) = -\eta$ for $w'_{t+1} := w_t - \eta g_t$. To see that $w_{t+1} = \Pi_W(w'_{t+1}) = w'_{t+1}$, note that by assumption $\eta \leq 1/\sqrt{n}$, hence

$$\left\|w'_{t+1}\right\|^2 = \sum_{i=1}^{2d} w'_{t+1}(i)^2 \leq \sum_{t=1}^{t} w'_{t+1}(i_t)^2 + d(n\epsilon\eta)^2 = \eta^2(t + n/d) \leq (t + 1)/n,$$

and $w'_{t+1}$ remains inside $W$ for all $t < n$. Finally, since the desired event occurs with probability $1/2$ by Lemma 2, we are done. $\qquad \square$

## B.1 Lower bound for large step sizes

When the step size is large, the projections actually alleviate the problematic nature of our construction, to the point where they can be exploited to obtain any convergence rate with the full iterate averaging. Notably though, concatenating our construction with a standard lower bound (e.g., Lemma 14) the best convergence rate possible is $n^{-1/4}$ with $\eta = n^{-1/4}$ which is the same as what would be achieved by the somewhat more reasonable choice of $\eta = n^{-3/4}$ that does not rely on the projections.

**Theorem 9** (Large step size case of Theorem 1). *Let $n \in \mathbb{N}$, $n \geq 4$, $d \geq 2^{4n \log n}$, and $W = \mathcal{B}_0^{2d}(1)$. Then there exists a distribution over instance set $Z$ and a 4-Lipschitz convex loss function $f : W \times Z \to \mathbb{R}$ such that*

  *(i) the optimization error is large; $\mathbb{E}_{S \sim \mathcal{Z}^n}\left[\widehat{F}(\widehat{w}_S) - \widehat{F}(w_S^\star)\right] = \Omega\left(\frac{1}{\eta\sqrt{n}}\right)$,*

  *(ii) the generalization gap is large; $\mathbb{E}_{S \sim \mathcal{Z}^n}\left[\widehat{F}(\widehat{w}_S) - F(\widehat{w}_S)\right] = \Omega\left(\frac{1}{\eta\sqrt{n}}\right)$,*

*where $\widehat{w}$ is any suffix average of SGD with step size $\eta > 1/\sqrt{n}$.*

*Proof.* The analysis parts ways from the small step size case after Lemma 2. Instead of Lemma 3, we make the claim below.

*Lemma 4. For all $\tau \in [n]$, it holds that $t < \tau \implies w_\tau(i_t) \leq -\eta(1+\eta^2)^{t-\tau}$, where $i_t := I(w_t)$ and $\tau \leq n+1$. In addition, for any suffix average $\widehat{w}$, it holds that*

$$\sum_{t=n/4}^{n} |\widehat{w}(i_t)| \geq \frac{1}{20\eta}.$$

The important consequence of the above lemma is that whichever suffix average we take, we will end up with an $\Omega(1/\eta)$ mass in the total bad coordinate summation. We now show this translates to an empirical risk lower bound as claimed. Ignoring the negligible contribution of $\epsilon$, by Lemma 15 we have;

$$\widehat{F}(w) = \frac{1}{n}\sum_{s=1}^{n} f(w; z_s) \geq \frac{1}{n}\sum_{s=1}^{n}\sqrt{\sum_{t=s+1}^{n} w(i_t)^2} \geq \frac{1}{5\sqrt{n}}\sum_{t=n/4}^{n} |w(i_t)| \geq \frac{1}{100\eta\sqrt{n}},$$

where the last inequality follows from Lemma 4. This completes the proof. $\qquad\square$

*Proof of Lemma 4.* For $t \in [n]$, denote $w_t' := w_t - \eta g(w_t, z_t)$ so that now $w_{t+1} \leftarrow \Pi_W(w_t')$. Informally, we have $\|w_t'\|^2 \leq 1 + \eta^2$ for all $t$, when we ignore the negligible $\epsilon$ component. Formally, let

$$\zeta_t(i) = \begin{cases} 0 & i \leq d, \\ -\epsilon z_t(i) & i > d, \end{cases}$$

so that $g(w_t; z_t) = \zeta_t + e_{i_t} + e_{(i_t+d)}$, and observe

$$\begin{aligned} \|w_{t+1}'\|^2 &= \|w_t - \eta\zeta_t - \eta e_{i_t} - \eta e_{(t_t+d)}\|^2 \\ &\leq 1 + dn\eta^2\epsilon^2 + 2\eta^2 \\ &\leq 1 + (n/d)\eta^2 + 2\eta^2 \\ &\leq 1 + 3\eta^2. \end{aligned}$$

Now, set $\gamma := 3\eta^2$, let $t < \tau \in [n]$, and observe;

$$w_\tau(i_t) = \Pi_W(w_\tau')(i_t) = \frac{w_\tau'(i_t)}{\|w_\tau'\|} \leq w_\tau'(i_t)(1+\gamma)^{-1}$$

where the inequality follows from the norm bound and since $w_l'(i_t) \leq 0$ for all $l \in [d]$. In addition, for $\tau - 1 > t$, we have $w_\tau'(i_t) = w_{\tau-1}(i_t)$, hence

$$w_\tau'(i_t)(1+\gamma)^{-1} = w_{\tau-1}(i_t)(1+\gamma)^{-1} \leq \cdots \leq w_{t+1}'(i_t)(1+\gamma)^{t-\tau} = -\eta(1+\gamma)^{t-\tau},$$

therefore,

$$t < \tau \implies w_\tau(i_t) \leq -\eta(1+\gamma)^{t-\tau}, \tag{21}$$

which proves the first part. For the second part, we begin by computing the values in each individual coordinate.

**The individual coordiantes $\widehat{w}(i_t)$.** let $\widehat{w}$ be the average of the last $k$ iterates $w_{n-k+2}, \ldots, w_{n+1}$, and set $\tau_0 := n - k$. Fix $t \in [n]$, set $l := \max(\tau_0 + 1, t + 1)$, and observe

$$|\widehat{w}(i_t)| \geq \frac{1}{k} \sum_{\tau=l}^{n+1} |w_\tau(i_t)| \geq \frac{\eta}{k} \sum_{\tau=l}^{n+1} (1 + \eta^2)^{t-\tau},$$

where the first inequality follows since all values are negative and the second from Eq. (21). We have

$$\sum_{\tau=l}^{n+1} (1 + \gamma)^{t-\tau} = \sum_{k=l-t}^{n+1-t} (1 + \gamma)^{-k}$$

$$= (1 + \gamma)^{-(l-t)} \left(1 - (1 + \gamma)^{-(n+2-l)}\right) \frac{1}{1 - (1 + \gamma)^{-1}}$$

$$= (1 + \gamma)^{t-l} \left(1 - (1 + \gamma)^{l-n-2}\right) \frac{(1 + \gamma)}{(1 + \gamma) - 1}$$

$$= (1 + \gamma)^{t-l+1} \left(1 - (1 + \gamma)^{l-n-2}\right) \frac{1}{\gamma}$$

$$=: (*).$$

Now,

$$t \leq \tau_0 \implies (*) = (1 + \gamma)^{t+k-n-1} \left(1 - (1 + \gamma)^{-k}\right) \frac{1}{\gamma}$$

$$\implies |\widehat{w}(i_t)| \geq \frac{\eta}{\gamma k} (1 + \gamma)^{t+k-n-1} \left(1 - (1 + \gamma)^{-k}\right) \tag{22}$$

$$\text{and } t > \tau_0 \implies (*) = \left(1 - (1 + \gamma)^{t-n-1}\right) \frac{1}{\gamma}$$

$$\implies |\widehat{w}(i_t)| \geq \frac{\eta}{\gamma k} \left(1 - (1 + \gamma)^{t-n-1}\right). \tag{23}$$

Before moving on to bound the sum of values in the coordinates, we record the following basic facts which will be used repeatedly. By Bernoulli's inequality and our assumption that $\eta > 1/\sqrt{n}$, we have $(1 + \gamma)^m = (1 + 3\eta^2)^m \geq 1 + 3\eta^2 m \geq 1 + 3m/n$. Hence,

$$(1 + \gamma)^{-m} = \frac{1}{(1 + \gamma)^m} \leq \frac{1}{1 + 3m/n}$$

$$\implies 1 - (1 + \gamma)^{-m} \geq \frac{3m/n}{1 + 3m/n},$$

and then,

$$m \geq n/4 \implies 1 - (1 + \gamma)^{-m} \geq (3/4)/4 \geq 1/6, \tag{24}$$

$$\text{and } \sum_{j=0}^{m} (1 + \gamma)^{-j} \geq \frac{1 - (1 + \gamma)^{-m}}{1 - (1 + \gamma)^{-1}} \geq \frac{1/6}{1 - \frac{1}{1+\gamma}} = \frac{1}{6\gamma}. \tag{25}$$

**Bounding the sum of coordinate values.** We first consider the case that $\tau_0 < n/2$;

$$\sum_{t=n/4}^{n} |\widehat{w}(i_t)| \geq \frac{\eta}{\gamma k} \sum_{t=n/2}^{3n/4} \left(1 - (1 + \eta^2)^{t-n-1}\right) \qquad \text{(by Eq. (23))}$$

$$\geq \frac{\eta}{\gamma k} \sum_{t=n/2}^{3n/4} \frac{1}{6} \qquad \text{(by Eq. (24))}$$

$$= \frac{\eta}{6\gamma k} (3n/4 - n/2)$$

$$= \frac{n\eta}{24\gamma k}$$

$$\geq \frac{n}{100\eta k} \geq \frac{1}{100\eta},$$

which proves the desired result (recall that $\gamma = 3\eta^2$). Assume now $\tau_0 \geq n/2$, and observe;

$$\sum_{t=\tau_0+1}^{n} |\widehat{w}(i_t)| \geq \frac{\eta}{\gamma k} \sum_{t=\tau_0+1}^{n} \left(1 - (1+\gamma)^{t-n-1}\right) \qquad \text{(by Eq. (23))}$$

$$= \frac{\eta}{\gamma k} \sum_{j=1}^{k-1} \left(1 - (1+\gamma)^{-j}\right)$$

$$= \frac{\eta}{\gamma k} \left(k - 1 - \sum_{j=1}^{k-1} (1+\gamma)^{-j}\right)$$

$$= \frac{\eta}{\gamma k} \left(k - 1 - (1+\gamma)^{-1} \frac{1 - (1+\gamma)^{1-k}}{1 - (1+\gamma)^{-1}}\right)$$

$$= \frac{\eta}{\gamma k} \left(k - 1 - \frac{1 - (1+\gamma)^{1-k}}{\gamma}\right)$$

$$= \frac{\eta}{\gamma^2 k} \left(\gamma(k-1) - (1 - (1+\gamma)^{1-k})\right)$$

$$= \frac{\eta}{\gamma^2 k} \left(\gamma(k-1) + (1+\gamma)^{1-k}\right). \qquad (26)$$

In addition,

$$\sum_{t=n/4}^{\tau_0} |\widehat{w}(i_t)| \geq \frac{\eta}{\gamma k} \left(1 - (1+\gamma)^{-k}\right) \sum_{t=n/4}^{\tau_0} (1+\gamma)^{t+k-n-1} \qquad \text{(by Eq. (22))}$$

$$= \frac{\eta}{\gamma k} \left(1 - (1+\gamma)^{-k}\right) \sum_{t=n/4}^{\tau_0} (1+\gamma)^{t-\tau_0} \qquad \text{(by Eq. (22))}$$

$$= \frac{\eta}{\gamma k} \left(1 - (1+\gamma)^{-k}\right) \sum_{j=0}^{\tau_0-n/4} (1+\gamma)^{-j}$$

$$\geq \frac{\eta}{\gamma k} \left(1 - (1+\gamma)^{-k}\right) \sum_{j=0}^{n/4} (1+\gamma)^{-j}$$

$$\geq \frac{\eta}{6\gamma^2 k} \left(1 - (1+\gamma)^{-k}\right), \qquad (27)$$

where in the last inequality we have applied Eq. (25). Now, combining Eq. (26) and Eq. (27), we obtain

$$\sum_{t=n/4}^{n} |\widehat{w}(i_t)| \geq \frac{\eta}{\gamma^2 k} \left(\gamma k - \gamma - 1 + (1+\gamma)^{1-k}\right) + \frac{\eta}{6\gamma^2 k} \left(1 - (1+\gamma)^{-k}\right)$$

$$\geq \frac{\eta}{6\gamma^2 k} \left(\gamma k - \gamma - 1 + (1+\gamma)^{1-k} + 1 - (1+\gamma)^{-k}\right)$$

$$= \frac{\eta}{6\gamma^2 k} \left(\gamma k - \gamma + (1+\gamma)^{1-k} - (1+\gamma)^{-k}\right)$$

$$\geq \frac{\eta}{6\gamma k} (k-1)$$

$$\geq \frac{\eta}{12\gamma}$$

$$= \frac{1}{36\eta},$$

which proves the desired result also in the second case, and completes the proof. □

## B.2 SCO with non convex components

*Proof of Theorem 2.* We define the distribution $\mathcal{Z} = \mathcal{Z}(\delta)$ over the set of datapoints $Z$ by

$$\forall i \le d; \quad z(i) = \begin{cases} -1 & \text{w.p. } \delta \\ 1 & \text{w.p. } \delta \\ 0 & \text{w.p. } (1 - 2\delta), \end{cases}$$

$$\forall i > d; \quad z(i) = \mathbb{1}\left\{z(i - d) = -1\right\}.$$

We consider the same loss function of Theorem 1, but leave the norm-like component without the square-root;

$$f(w; z) := \phi(w; z) + v_z(w), \quad v_z(w) := \sum_{i=1}^{d} z(i) w(i)^2,$$

where $\phi$ is defined as in Eq. (13). We also define the gradient oracle for $\phi$ as we have done in the convex case Eq. (15), Eq. (16), and Eq. (17), repeated here for convenience;

$$g_\phi(w; z)(i) := \begin{cases} \mathbb{1}\left\{i = I(w)\right\} & i \le d \\ -\epsilon z(i) + \mathbb{1}\left\{i = I(w) + d\right\} & i \ge d \end{cases},$$

$$I(w) := \min\left\{i \in [d] \mid i \in \arg\max_{1 \le j \le d}\{w(j) + w(j + d)\}\right\},$$

$$i_t := I(w_t).$$

Here, unlike the construction in Theorem 1 we need $\epsilon$ to depend on $\eta$, and set $\epsilon := \eta/d$. The next lemma establishes the SGD iterates end up "overfitting" the empirical objective, and follows from a proof that is essentially identical to Lemma 2 and Lemma 3. The only difference is that here the training examples have $-1$ rather than $1$ in the critical coordinates.

*Lemma 5. For $\delta = 1/4n^2$, we have with probability $\ge 1/2$ that for all $\tau \in [n]$ and $t < \tau$;*

- $w_\tau(i_t) = -\eta,$

- $1 \le s < t \implies z_s(i_t) = -1,$ *and*

- $t \le s \le n \implies z_s(i_t) \le 0.$

Thus, let $\tau \le n + 1$, and denote $\overline{w}_\tau := \frac{1}{n-\tau+2} \sum_{t=\tau}^{n+1} w_t$. By a derivation identical to that of the convex case, we obtain for all $1 \le t \le n/2$, $\overline{w}_\tau(i_t) \le -\frac{\eta}{2}$. Hence, for $s \le n/4$,

$$v_{z_s}(\overline{w}_\tau) \le - \sum_{t=n/4+1}^{n/2} \overline{w}_\tau(i_t)^2 = -\frac{n\eta^2}{16},$$

therefore

$$\widehat{F}(\overline{w}_\tau) = \frac{1}{n} \sum_{t=1}^{n} \phi(\overline{w}_\tau; z_t) + \frac{1}{n} \sum_{t=1}^{n} v_{z_t}(\overline{w}_\tau)$$

$$\le 2\epsilon\eta n - \frac{1}{n}\frac{n}{2}\frac{n\eta^2}{16}$$

$$\le -\frac{n\eta^2}{64}.$$

To conclude the proof, we note that

$$F(\overline{w}_\tau) \ge -\epsilon \sum_{i=d+1}^{2d} \delta(\eta\epsilon n) = -\epsilon^2 d\delta n\eta \ge -\eta^2/n,$$

and the result follows. $\qquad\square$

## B.3 SCO with strongly convex components

*Proof of Theorem 3.* We first make the argument for an unbounded domain, so that no projections take place. Let

$$f(w; z) := \phi(w; z) + v_z(w) + \frac{\lambda}{2} \|w\|^2,$$

where $\phi$ and $v$ are defined by

$$\phi(w; z) := -\epsilon \sum_{i=d+1}^{2d} z(i) w(i) + \max_{1 \leq i \leq d} \{w(i) + \epsilon w(i + d)\}, \tag{28}$$

$$v_z(w) := \sqrt{\sum_{i=1}^{d} z(i) w(i)^2}, \tag{29}$$

and $\epsilon := 1/d$. These are essentially the same definitions as in our main construction Eq. (13) and Eq. (14), but with an added $\epsilon$ factor inside the max component of $\phi$. This only makes our formal argument simpler, but otherwise does not make any significant difference. For the gradient oracle, we define

$$g_\phi(w; z)(i) := \begin{cases} \mathbb{1}\{i = I(w)\} & i \leq d \\ -\epsilon z(i) + \epsilon \mathbb{1}\{i = I(w) + d\} & i \geq d \end{cases}, \tag{30}$$

where $I(w) := \min \left\{ i \in [d] \mid i \in \arg\max_{1 \leq j \leq d} \{w(j) + \epsilon w(j + d)\} \right\}, \tag{31}$

and again denote the index picked by $g$ on round $t \in [n]$ by

$$i_t := I(w_t). \tag{32}$$

We then set

$$g(w; z) := g_\phi(w; z) + \nabla v_z(w) + \lambda w.$$

Clearly, for all $w, z \in \mathbb{R}^{2d}$, $g(w; z) \in \partial_w f(w; z)$. Following a direct computation, we get that

$$g(w; z)(i) = \begin{cases} \mathbb{1}\{i = I(w)\} + \frac{z(i) w(i)}{v_z(w)} + \lambda w(i) & i \leq d, \\ \epsilon \mathbb{1}\{i = I(w) + d\} - \epsilon z(i) + \lambda w(i) & i > d, \end{cases}$$

where $I$ is defined in Eq. (16). Hence, the stochastic gradient steps $w_{t+1} \leftarrow w_t - \eta_t g(w_t, z_t)$ are given by

$$w_{t+1}(i) = \begin{cases} \left(1 - \eta_t (z_t(i) v_{z_t}(w)^{-1} + \lambda)\right) w_t(i) - \eta_t \mathbb{1}\{i = I(w)\} & i \leq d \\ (1 - \eta_t \lambda) w_t(i) + \epsilon \eta_t z_t(i) - \eta_t \epsilon \mathbb{1}\{i = I(w) + d\} & i > d. \end{cases} \tag{33}$$

The next lemma makes a similar assertion as Lemma 3 and follows from similar arguments.

*Lemma* 6. *With probability* $\geq 1/2$, *for all* $\tau \in [n]$ *we have*

$$w_{\tau+1}(i) = \begin{cases} -\eta_t \prod_{s=t+1}^{\tau} (1 - \eta_s \lambda) & i = i_t, t \in [\tau], \\ 0 & i \in [d] \setminus \{i_1, \dots i_\tau\}, \end{cases}$$

*where* $z_1(i_t) = \dots = z_{t-1}(i_t) = 1$, *and* $z_t(i_t) = \dots = z_n(i_t) = 0$.

Next, a simple derivation shows the empirical risk is large for any suffix average. Thus, let $\tau \leq n + 1$, and denote $\overline{w}_\tau := \frac{1}{n-\tau+2} \sum_{t=\tau}^{n+1} w_t$. By Lemma 6, assuming the event from the lemma occurs we have

$$t \leq n/2 \implies |\overline{w}_\tau(i_t)| \geq \frac{1}{2} |w_{n+1}(i_t)| = \frac{\eta_t}{2} \prod_{s=t+1}^{n} (1 - \eta_s \lambda).$$

Therefore, noting that $\phi(\overline{w}_\tau; z_s) \geq -\epsilon^2 dn/\lambda \geq -\epsilon n/\lambda \geq -1/(\lambda n)$, by Lemma 15 we obtain;

$$\widehat{F}(\overline{w}_{\tau:n}) \geq \frac{\lambda}{2} \|\overline{w}_{\tau:n}\|^2 - \frac{1}{\lambda n} + \frac{1}{10\sqrt{n}} \sum_{t=n/4}^{n/2} \eta_t \prod_{s=t+1}^{n} (1 - \eta_s \lambda)$$

$$\geq \frac{\lambda}{2} \|\overline{w}_{\tau:n}\|^2 - \frac{1}{\lambda n} + \frac{1}{40\lambda \sqrt{n}}. \qquad \text{(for } \eta_t = 1/\lambda t)$$

Noting that $\widehat{F}(0) = 0$, we obtain the claim on the optimization error. For the generalization gap, first note that for any $t \leq n + 1$,

$$\mathbb{E}_z \phi(w_t; z) \leq -\epsilon \sum_{i=d+1}^{2d} \delta w_t(i) + \epsilon n/\lambda$$
$$\leq \epsilon \delta n/\lambda + \epsilon n/\lambda$$
$$\leq 2\epsilon n/\lambda,$$

and observe;

$$F(\overline{w}_\tau) \leq \frac{2\epsilon n}{\lambda} + \frac{\lambda}{2} \|\overline{w}_\tau\|^2 + \mathbb{E}_{z \sim \mathfrak{z}} \left[ \sqrt{\sum_{t=1}^{n} z(i_t)\overline{w}_\tau(i_t)^2} \right]$$

$$\leq \frac{2\epsilon n}{\lambda} + \frac{\lambda}{2} \|\overline{w}_\tau\|^2 + \sqrt{\delta \sum_{t=1}^{n} \overline{w}_\tau(i_t)^2} \qquad \text{(by Jensen's inequality)}$$

$$\leq \frac{2\epsilon n}{\lambda} + \frac{\lambda}{2} \|\overline{w}_\tau\|^2 + \frac{1}{2n} \sum_{t=1}^{n} |\overline{w}_\tau(i_t)|$$

$$\leq \frac{2\epsilon n}{\lambda} + \frac{\lambda}{2} \|\overline{w}_\tau\|^2 + \frac{\log n}{2\lambda n} \qquad \text{(for } \eta_t = 1/\lambda t)$$

$$\leq \frac{\lambda}{2} \|\overline{w}_\tau\|^2 + \frac{\log n}{\lambda n}. \qquad \text{(since } 2\epsilon n/\lambda \leq \log n/(2\lambda n))$$

Combining the inequalities in the last two displays completes the proof.

**Bounded domain case with $\lambda \geq 1/\sqrt{n}$.** In this case projections happen, but owed to our assumption on $\lambda$ we will see their effect is negligible. Denote $w'_s := w_s - \eta g(w_s, z_s)$ so that now $w_{s+1} \leftarrow \Pi_W(w'_s)$. Observe that by Eq. (33) under the event of Lemma 6, we have

$$w'_{\tau+1}(i) = \begin{cases} 0 & i \in [d] \setminus \{i_1, \ldots, i_\tau\}, \\ (1 - \eta_\tau \lambda)w_\tau(i) + \eta_\tau \mathbb{1}\{i = i_\tau\} & i \in \{i_1, \ldots, i_\tau\}, \\ (1 - \eta_\tau \lambda)w_\tau(i) + \epsilon \eta_\tau z_t(i)(-1 + \mathbb{1}\{i = i_\tau + d\}) & i \in [d+1, 2d], \end{cases}$$

thus

$$w'_{\tau+1}(i)^2 \leq \begin{cases} 0 & i \in [d] \setminus \{i_1, \ldots, i_\tau\}, \\ (1 - \eta_\tau \lambda)^2 w_\tau(i)^2 & i \in \{i_1, \ldots, i_{\tau-1}\}, \\ \eta_\tau^2 & i = i_\tau, \\ (n\epsilon/\lambda)^2 & i \in [d+1, 2d]. \end{cases}$$

Note that by our assumption that $\lambda \geq 1/\sqrt{n}$ and our choice of $\epsilon = 1/d$, we have $n^2\epsilon^2/\lambda^2 \leq n/d^2$. Now, fix $\tau \geq 2n/3$, and observe;

$$\left\|w'_{\tau+1}\right\|^2 \leq \eta_\tau^2 + d(n/d^2) + (1 - \eta_\tau \lambda)^2 \sum_{i=1}^{2d} w_\tau(i)^2$$

$$= \eta_\tau^2 + n/d + (1 - \eta_\tau \lambda)^2 \|w_\tau\|^2$$

$$\leq \eta_\tau^2 + n/d + (1 - \eta_\tau \lambda)^2$$

$$= \frac{1}{\lambda^2 \tau^2} + \frac{n}{d} - \frac{2}{\tau} + \frac{1}{\tau^2}$$

$$\leq \frac{2}{\lambda^2 n \tau} + 1 + \frac{n}{d} + \frac{9}{4n^2} - \frac{2}{\tau}$$

$$\leq \frac{3}{2\tau} + 1 + \frac{10}{4n^2} - \frac{2}{\tau}$$

$$\leq 1.$$

In the above, we have used that $n/d \leq 1/4n^2$, and that $1/2\tau \geq 1/(2n) \geq 10/4n^2$ for $n \geq 10$. Hence, from round $2n/3$ onwards projections do not occur anymore. To conclude the proof, we note that we can lower bound the empirical loss precisely as we did before but over rounds $4n/6$ to $5n/6$, rather than $n/4$ to $n/2$. In addition, the population loss has only improved since the per coordinate values in all iterates have only decreased in magnitude as a result of the projections. $\qquad\square$

*Proof of Lemma 6.* Note that for all $t \in [n]$ and any $i \in [d+1, 2d] \setminus \{i_1 + d, \ldots, i_t + d\}$, by Eq. (33) we have

$$
\begin{aligned}
w_{t+1}(i) &= (1 - \eta_t \lambda) \, w_t(i) + \epsilon \eta_t z_t(i) \\
&= \sum_{s=1}^{t} \epsilon \eta_s z_s(i) \prod_{l=s+1}^{t} (1 - \eta_l \lambda) \, .
\end{aligned}
\tag{34}
$$

Hence, the number of times $z_s(i) = 1$ for $s \leq t$ determines the maximality of $w_{t+1}(i)$. In other words, the extra component in the gradient update effects all coordinates equally, and the situation here is no different than the convex case. Thus, by Lemma 2 and the same arguments as given in Lemma 3, we have that with probability $\geq 1/2$, for all $t \in [n]$, $i_t = I(w_t) = J_t$. Therefore, by Eq. (33);

$$
w_{t+1}(i) = \begin{cases}
-\eta_t & i = i_t; \\
-\eta_s \prod_{l=s+1}^{t}(1 - \eta_l \lambda) & i = i_s, \ s < t; \\
0 & i \in [d] \setminus \{i_1, \ldots, i_t\} \, .
\end{cases}
$$

Note that the $z_t(i) v_{z_t}(w)^{-1}$ component in Eq. (33) does not contribute since $z_t(i_s) = 0$ for all $t \geq s$, given our event. $\qquad\square$

## C  Proofs for Section 4

In what follows we provide the standard analysis of SGD with the iterate averaging scheme specified in Theorem 4. The theorem stated and proved below provides the rate of convergence on the target objective function from which gradients are sampled (as similar analyses normally do); note that we use it in the context where the target objective is the *empirical* loss given by the training set. This should be contrasted with the goal of Theorem 4, which is to establish the convergence rate on the *population* objective. Our only motivation for proving the below theorem is to argue the generalization gap upper bound established in Corollary 2.

**Theorem 10.** *Let $W \subset \mathbb{R}^d$ with diameter $D$, and $f_1, \ldots, f_n$ be a sequence of convex, $G$-Lipschitz losses sampled i.i.d. from some distribution $\mathcal{F}$. Further, let $w^\star := \min_{w \in W} \mathbb{E}_{f \sim \mathcal{F}} f(w)$ denote the minimizer of the expected function. Then, the weighted average $\overline{w} := \frac{2}{n+1} \sum_{t=1}^{n} \frac{n-t+1}{n} w_t$ of the iterates produced by SGD with step size $\eta = \frac{D}{G\sqrt{n}}$ obtains the following upper bound:*

$$
\mathbb{E}_{f_1, \ldots, f_n, f \sim \mathcal{F}} \left[ f(\overline{w}) - f(w^\star) \right] \leq \frac{4GD}{\sqrt{n}} \, .
$$

*Proof.* Observe,

$$
\begin{aligned}
\mathbb{E}\left[ f(\overline{w}) - f(w^\star) \right] &\leq \frac{2}{n+1} \sum_{t=1}^{n} \frac{n-t+1}{n} \mathbb{E}\left[ f(w_t) - f(w^\star) \right] \\
&= \frac{2}{n+1} \mathbb{E}\left[ \sum_{t=1}^{n} \frac{n-t+1}{n} \left( f_t(w_t) - f_t(w^\star) \right) \right] .
\end{aligned}
\tag{35}
$$

By the standard SGD analysis,

$$
f_t(w_t) - f_t(w^\star) \leq \nabla f_t(w_t)^\mathsf{T}(w_t - w^\star) \leq \frac{1}{2\eta}\left( D_t^2 - D_{t+1}^2 \right) + \frac{\eta}{2} G^2,
$$

where $D_t := \|w_t - w^\star\|$. Now,

$$\sum_{t=1}^{n} \frac{n-t+1}{n}\left(f_t(w_t) - f_t(w^\star)\right) \leq \frac{1}{2\eta}\sum_{t=1}^{n} \frac{n-t+1}{n}\left(D_t^2 - D_{t+1}^2\right) + \frac{\eta G^2 n}{2}$$

$$= \frac{D_1^2}{2\eta} + \frac{1}{2\eta}\sum_{t=2}^{n} D_t^2 \left(\frac{n-t+1}{n} - \frac{n-t}{n}\right) + \frac{\eta G^2 n}{2}$$

$$= \frac{D_1^2}{2\eta} + \frac{1}{2\eta n}\sum_{t=2}^{n} D_t^2 + \frac{\eta G^2 n}{2}$$

$$\leq \frac{D^2}{\eta} + \frac{\eta G^2 n}{2}$$

$$\leq 2GD\sqrt{n}.$$

Plugging the above inequality into Eq. (35), we obtain

$$\mathbb{E}\left[f(\overline{w}) - f(w^\star)\right] \leq \frac{4GD}{\sqrt{n}},$$

which completes the proof. □

## D  Proofs for Section 5

### D.1  Lower bound for multi-epoch SGD

*Proof of Theorem 5.* First, note that without modifications, the strategy of Theorem 1 breaks after the first epoch; it will just keep pointing the gradient on coordinates with an all ones sequence. We use the idea we can fully "record" into the iterate $w_t^k$ the precise samples we have stepped through so far, and define a gradient oracle that will cause the iterate to advance on fresh bad coordinates in every new epoch. We will work with the datapoints set $Z = \{0,1\}^d$ and define the distribution $\mathcal{Z} = \mathcal{Z}(\delta)$ over $Z$ by letting $z(i) \sim \mathrm{Ber}(\delta)$ for all $i \in [d]$. We consider two separate portions of a vector $w \in \mathbb{R}^{d'}$, which we denote by $w[\cdot;0] \in \mathbb{R}^d$ and $w[\cdot,\cdot;1] \in \mathbb{R}^{d \times nK}$. The first portion with $d$ entries is where the bad gradient steps will be made and where we will eventually suffer the loss from. The second consists of $dnK$ entries and is used to encode the samples observed during the optimization process. Our loss function is defined as follows;

$$v_z(w) := \sqrt{\sum_{i=1}^{d} z(i)w[i;0]^2},$$

$$\phi(w;z) := \epsilon \sum_{i=1}^{d} (1+z(i))\max_{j\in[nK]}\{w[i,j;1]\} + \max_{i\in[d]}\{w[i;0])\}, \tag{36}$$

$$f(w;z) := \phi(w;z) + v_z(w).$$

Again, we choose $\epsilon > 0$ sufficiently small so that the loss induced by it is negligible, and so that $f$ is 4-Lipschitz. The gradient oracle we use is specified by that of the $\phi$ function;

$$g_\phi(w;z)[i,j;1] = \epsilon(1+z(i))\mathbb{1}\{j = I_1(w[i,\cdot;1])\}$$

$$g_\phi(w;z)[\cdot;0] = e_{I_0(w)}$$

$$\text{where } I_1(x) := \min\left\{j \in [nK] \,\Big|\, j \in \arg\max_{l\in[nK]}\{x(l)\}\right\} \quad \text{for } x \in \mathbb{R}^{nK},$$

$$I_0(w) := \min\left\{i \in [d] \,\Big|\, i \in \arg\max_{l\in[d]}\{w[l;0]\}, \text{ and } i \in \arg\min_{l\in[d]}\left\{V(w)(l)\right\}\right\},$$

$$\text{and } V(w) := \sum_{j=t_0(w)}^{nK} w[\cdot,j;1],$$

where $t_0(w)$ denotes the first global iteration index of the current epoch. This index is easy to infer from $w[\cdot, \cdot; 1]$ since the $t$'th SGD iteration in epoch $k$ results in values strictly smaller than 0 in all entries of $w[\cdot, \tau; 1]$, where $\tau = n(k-1) + t$ (and 0 remains from initialization for $\tau' > \tau$). In words, we design our "adversarial" gradient oracle so that it will choose the coordinate for $\max_{i \in [d]} \{w[i; 0]\}$, by "looking" in $w[\cdot, \cdot; 1]$ and choosing the coordinate $i_t^k := I_0(w_t^k) \in [d]$ such that the number if times $z_s^k(i_t^k) = 1$ for $s < t$ is largest. An illustration is provided in Fig. 1.

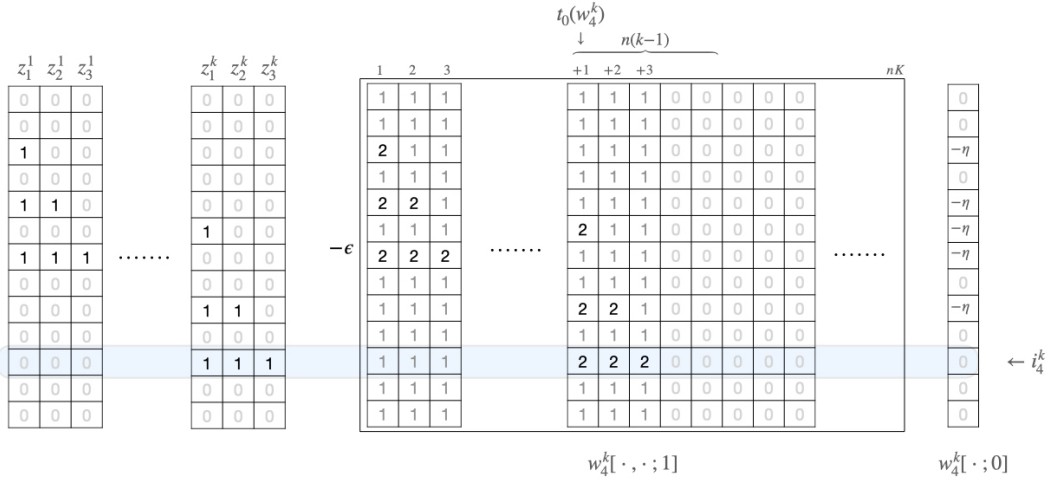

Figure 1: Illustration of gradient oracle mechanism

In similar spirit to the the basic construction from Theorem 1, we will ensure that with high probability, the coordinates selected by our gradient oracle are such that $z_s^k(i_t^k) = 1$ for all $s < t$, and $z_s^k(i_t^k) = 0$ for all $s \geq t$. To that end, we first assert the existence of a set of datapoints $Z \subset \{0, 1\}^d$ where a desired property described next holds with sufficiently high probability. Consider some arbitrary ordered set $S = \{z_1, \ldots, z_n\} \subset \{0, 1\}^d$. For $t \in [n]$, denote

$$E^{(1,t)} := \left\{ i \in [d] \mid z_s(i) = 1 \; \forall s < t \right\},$$

$$E^{(1,t;K)} := \left\{ J_t^1, \ldots, J_t^K \mid J_t^k \text{ is the k'th smallest} \in E^{(1,t)} \right\},$$

$$E^{(t,0)} := \left\{ i \in [d] \mid z_s(i) = 0 \; \forall s \geq t \right\}.$$

So $E^{(1,t;K)}$ is just the first $K$ elements of $E^{(1,t)}$, where we enumerate the coordinates by the superscript in increasing order. We say the event $\mathcal{E}$ holds for the set $S$, or equivalently that $S \in \mathcal{E}$ if for all $t \in [n]$, we have $|E^{(1,t;K)}| = K$, and $E^{(1,t;K)} \subseteq E^{(t,0)}$. In words, $S \in \mathcal{E}$ if for every $t \in [n]$, the first $K$ coordinates $\{J_t^1, \ldots, J_t^K\}$ that have a prefix of $(t-1)$ ones; $s < t \implies z_s(J_t^k) = 1$, also satisfy that they have a suffix of zeros; $s \geq t \implies z_s(J_t^k) = 0$.

*Lemma 7. There exists a set of datapoints $Z = \{\zeta_1, \ldots, \zeta_n\} \subset \{0, 1\}^d$, such that*

$$\Pr_{\pi_1, \ldots, \pi_K \sim \Pi([n])} \left( \forall k \in [K], \; \{z_1^k, \ldots, z_n^k\} \in \mathcal{E} \right) \geq 1/2,$$

*where $\pi_k$ is sampled by either single-shuffle or multi-shuffle, and $z_t^k := \zeta_{\pi_k(t)}$.*

With Lemma 7 in place, we can be sure bad coordinate sets will turn up in every epoch. Let $k \in [K]$ and $t \in [n]$, and assume the event from the lemma occurs. By the definition of our gradient oracle, it is bound to select one of the first $k$ coordinates that have a prefix of all ones, which we are assured by the lemma will also have a suffix of zeros. Formally, we argue by induction on $k, t$. The base case follows from the definition of $g_\phi$ and our assumption that the event occurs. For the inductive step, assume the selected coordinates $i_{t'}^{k'}$ of all prior rounds satisfy the inductive hypothesis. Then at most $k - 1$ of the first coordinates in $E^{(1,t,K)}$ could have been selected previously, since the inductive hypothesis implies every selected coordinate $i_{t'}^{k'}$ has exactly $t' - 1$ ones. Hence, the gradient oracle

will choose a coordinate from $E^{(1,t,K)}$, the elements of which are coordinates that also enjoy a suffix of zeros, as assured by the event from the lemma. In addition, note that our initialization at $w_1^1 = 0$ and our assumption that $\eta \le 1/\sqrt{2nK}$ (and that $\epsilon$ is negligibly small) ensure the iterate never leaves the domain $W$ thus no projections occur. Summarizing, we have that for every $k \in [K], t \le n+1$, it holds that:

$$t < t' \implies w_{t'}^k(i_t^k) = -\eta$$
$$\text{and } s < t \implies z_s^k(i_t^k) = 1.$$

To complete the proof, we will now prove a lower bound of $\Omega(\eta\sqrt{n/J})$ for the average iterate of the last $J$ epochs. The other terms in the bounds of the theorem statement follow from concatenating our problem instance dimension-wise with standard constructions — see Lemma 14. Proceeding, we slightly overload notation and denote $w(i)$ for $w[i,0]$. Let $\widehat{w}$ be the average of the iterates in the last $J \in [K]$ epochs;

$$\widehat{w} := \frac{1}{nJ} \sum_{k=K-J+1}^{K} \sum_{t=1}^{n+1} w_t^k.$$

For all $n/4 \le t \le 3n/4$ and $K - J + 1 \le k \le K$, we have

$$\left|\widehat{w}(i_t^k)\right| \ge \frac{1}{nJ} \sum_{t'=3n/4}^{n} \left|w_{t'}^k(i_t^k)\right| \ge \frac{\eta}{nJ}(n/4) = \frac{\eta}{4J}, \tag{37}$$

since $w_{t'}^k(i_t^k) = -\eta$ for all $t' \ge t$.

**Single-shuffle case.** Ignoring the negligible $\epsilon$ terms, we now have

$$1 \le s \le n/4 \implies f(\widehat{w}; z_s^1) \ge \sqrt{\sum_{k=1}^{K} \sum_{t=s}^{n/2} z_s^1(i_t^k)\widehat{w}(i_t^k)^2}$$

$$\ge \sqrt{\sum_{k=K-J+1}^{K} \sum_{t=n/4+1}^{n/2} z_s^1(i_t^k)\widehat{w}(i_t^k)^2} = \sqrt{(nJ/4)\frac{\eta^2}{16J^2}} = \sqrt{\frac{n}{64J}}\eta,$$

since $z_s^k(i_t^k) = z_s^1(i_t^k) = 1 \; \forall s < t, \; k \in [K]$. Therefore

$$F(\widehat{w}) = \frac{1}{n} \sum_{s=1}^{n} f(\widehat{w}; z_s^1) \ge \frac{n/4}{n}\sqrt{\frac{n}{64J}}\eta \ge \sqrt{\frac{n}{J}}\frac{\eta}{32},$$

concluding the proof for this case.

**Multi-shuffle case.** For $\zeta_i \in Z$ denote

$$I^k(\zeta_i) := \mathbb{1}\left\{z_t = \zeta_i, t \in [n/4, 3n/4]\right\},$$

$$I(\zeta_i) := \sum_{k=K-J+1}^{K} I^k(\zeta_i),$$

$$\widetilde{Z} := \left\{\zeta_i \in Z \mid I(\zeta_i) \ge J/4\right\}.$$

We wish to lower bound the size of $\widetilde{Z}$, to show that enough $\zeta_i$'s where incident in the $[n/4, 3n/4]$ iteration range in a sufficiently large number of epochs. (Our interest in this range stems from the desire to apply Eq. (37).) Observe;

$$\sum_{k=K-J+1}^{K} \sum_{i=1}^{n} I^k(\zeta_i) = J\frac{n}{2}$$

$$\implies \sum_{i=1}^{n} I(\zeta_i) = \frac{nJ}{2}.$$

Since $I(\zeta_i) \leq J$ for all $i \in [n]$, by the pegionhole principle we get that $|\widetilde{Z}| \geq n/4$. Otherwise, we would have

$$\sum_{i=1}^{n} I(\zeta_i) \leq |\widetilde{Z}|J + (n - |\widetilde{Z}|)\frac{J}{4} < J\frac{n}{4} + \frac{J}{4}n = \frac{nJ}{2}.$$

Now,

$$
\begin{aligned}
F(\widehat{w}) &\geq \frac{1}{n} \sum_{z \in \widetilde{Z}} f(\widehat{w}; z) \geq \frac{1}{n} \sum_{z \in \widetilde{Z}} \sqrt{\sum_{k=K-J+1}^{n} I^k(z) \sum_{t=3n/4}^{n} \widehat{w}(i_t^k)^2} \\
&\geq \frac{1}{n} \sum_{z \in \widetilde{Z}} \sqrt{\frac{J}{4} \sum_{t=3n/4}^{n} \frac{\eta^2}{16J^2}} \\
&= \frac{1}{n} \sum_{z \in \widetilde{Z}} \sqrt{\frac{n\eta^2}{16^2 J}} \\
&\geq \frac{\eta\sqrt{n}}{64\sqrt{J}},
\end{aligned}
$$

which concludes the multi-shuffle case and the proof as a whole. $\qquad\square$

*Proof of Lemma 7.* We will make our argument for an i.i.d. sampled instance set $Z$, and convert it to the stated result as follows. Assume $\mathcal{Z}$ is a distribution over $\{0, 1\}^d$ for which we establish the following;

$$\Pr_{Z \sim \mathcal{Z}^n}(Z \in \mathcal{E}) \geq \frac{1}{2^{1/K}}. \tag{38}$$

Clearly, applying a random permuation on $Z \sim \mathcal{Z}^n$ does not change its distribution, therefore

$$
\begin{aligned}
\frac{1}{2^{1/K}} \leq \Pr_{Z \sim \mathcal{Z}^n}(Z \in \mathcal{E}) &= \Pr_{Z \sim \mathcal{Z}^n, \pi \sim \Pi([n])}(\pi(Z) \in \mathcal{E}) \\
&= \sum_{Z \in Z^n} \Pr_{\mathcal{Z}^n}(Z)\Pr_{\pi \sim \Pi([n])}(\pi(Z) \in \mathcal{E}) \\
&\leq \max_{Z \in Z^n} \left\{ \Pr_{\pi \sim \Pi([n])}(\pi(Z) \in \mathcal{E}) \right\}.
\end{aligned}
$$

The above derivation implies the existence of $Z^\star \in Z^n$ with the property that

$$\Pr_{\pi_1, \ldots, \pi_K \sim \Pi([n])}(\forall k \leq K, \ \pi_k(Z^\star) \in \mathcal{E}) \geq \left(\frac{1}{2^{1/K}}\right)^K = \frac{1}{2}, \qquad\text{(multi-shuffle)}$$

and $\Pr_{\pi_1 \sim \Pi([n])}(\forall k \leq K, \ \pi_k(Z^\star) \in \mathcal{E}) \geq \frac{1}{2^{1/K}} \geq \frac{1}{2}, \quad$ where $\pi_1 = \ldots = \pi_K$. $\quad$(single-shuffle)

Therefore, for the rest of the proof we focus on proving the distribution $\mathcal{Z}$ as defined next satisfies the desired property Eq. (38). Let $\delta > 0$ which will be chosen in hindsight, and consider $\mathcal{Z} = \mathcal{Z}(\delta)$ where $z(i) \sim \mathrm{Ber}(\delta)$ for each $i \in [d]$ independently. Fix $t \in [n]$, and let $\mathcal{E}_t$ denote the event that $|E^{(1,t)}| \leq K$, and $E^{(1,t;K)} \subseteq E^{(t,0)}$. We will prove $\mathcal{E}_t$ holds with sufficiently high probability, so that $\mathcal{E} = \cap_{t \in [n]} \mathcal{E}_t$ holds w.p. $\geq 1/2^{1/K}$. Proceeding, assuming we choose $\delta$ and $d$ so that $K < d\delta^n$, by Hoeffding's inequality we have that

$$
\begin{aligned}
\Pr\left(|E^{(1,t)}| \leq K\right) &= \Pr\left(\sum_{i=1}^{d} \mathbb{1}\left\{i \in E^{(1,t)}\right\} \leq K\right) \\
&= \Pr\left(\sum_{i=1}^{d} \mathbb{1}\left\{i \in E^{(1,t)}\right\} - d\delta^{t-1} \leq K - d\delta^{t-1}\right) \\
&= \Pr\left(d\delta^{t-1} - \sum_{i=1}^{d} \mathbb{1}\left\{i \in E^{(1,t)}\right\} \geq d\delta^{t-1} - K\right) \\
&\leq e^{-(d\delta^{t-1}-K)^2/d} \leq e^{-(d\delta^n-K)^2/d}. \tag{39}
\end{aligned}
$$

In addition, for $i \in \{J_t^1, \ldots, J_t^K\} = E^{(1,t;K)}$, we have

$$\Pr(i \in E^{(t,0)}) = \Pr(\forall s \geq t, \; z_s(i) = 0) \geq (1 - \delta)^n.$$

Therefore,

$$\Pr(E^{(1,t;K)} \subseteq E^{(t,0)}) \geq (1 - \delta)^{nK}.$$

From the above and Eq. (39) we obtain

$$\Pr(\text{not } \mathcal{E}_t) = \Pr\left(E^{(1,t;K)} \not\subseteq E^{(t,0)} \text{ or } |E^{(1,t)}| < K\right) \leq e^{-(d\delta^n - K)^2/d} + 1 - (1 - \delta)^{nK},$$

hence,

$$\Pr\left(\cap_{t \in [n]} \mathcal{E}_t\right) \geq 1 - n(e^{-(d\delta^n - K)^2/d} + 1 - (1 - \delta)^{nK}). \tag{40}$$

To finish the proof, we choose $\delta$ and $d$ as follows. Set $\delta := 1/cn^2 K$, and note that

$$(1 - \delta)^{nK} = \left(1 - \frac{1}{cn^2 K}\right)^{nK} \geq 1 - \frac{1}{cn} \implies 1 - (1 - \delta)^{nK} \leq \frac{1}{cn}.$$

In addition, note that $-(d\delta^n - K)^2/d \leq -d\delta^{2n} + 2K\delta^n$, hence

$$e^{-(d\delta^n - K)^2/d} \leq \frac{1}{cn}$$
$$\impliedby e^{-d\delta^{2n} + 2K\delta^n} \leq \frac{1}{cn}$$
$$\iff \log(cn) + 2K\delta^n \leq d\delta^{2n}$$
$$\iff (\log(cn) + 2K\delta^n)(cn^2 K)^{2n} \leq d,$$

which holds for any $d \geq 2^{6n \log(cnK)} = (cnK)^{6n} \geq 2\log(cn)(cn^2 K)^{2n}$. Back to Eq. (40) we obtain for $c = \frac{4}{2^{1/K} - 1}$;

$$\Pr\left(\cap_{t \in [n]} \mathcal{E}_t\right) \geq 1 - n\left(\frac{1}{cn} + \frac{1}{cn}\right) = 1 - \frac{2}{c} \geq \frac{1}{2^{1/K}},$$

and we are done. $\qquad\qquad\square$

### D.2 Upper bound for single-shuffle multi-epoch SGD

First, we slightly generalize the notion of uniform argument stability and prove some supporting lemmas. We extend the definition of uniform-argument-stability Eq. (4) to one that enables more than one difference in the training sets. We give the definition below in notation suitable for SGD and the lemmas that follow;

$$\epsilon_{\text{stab}}^{\text{SGD}}(\tau; J) := \max_{f_1, \ldots, f_\tau, f_1', \ldots, f_J'} \max_{i_1, \ldots, i_J \in [\tau]} \left\| w_{\tau+1} - w_{\tau+1}' \right\|, \tag{41}$$

where $w_{\tau+1}'$ is the output of GD after swapping $f_1, \ldots, f_\tau$ in locations $i_1, \ldots, i_J$ with the other losses $f_1', \ldots, f_J'$. Lemma 8 given next generalizes Lemma 10 for the stability notion we have introduced above. The proof provided below is based on similar lemmas given in [20].

**Lemma 8.** *Let* $\{f(w; t)\}_{t=1}^n$ *be a set of n, G-Lipschitz losses, and* $F(w) = \frac{1}{n}\sum_{t=1}^n f(w; t)$. *Then, for a uniformly random permutation* $\pi: [n] \leftrightarrow [n]$, $f_t^k = f(\cdot; \pi(t)) \forall k$, *it holds that for single-shuffle SGD;*

$$\mathbb{E}\left[F(w_t^k) - f_t^k(w_t^k)\right] \leq G\epsilon_{\text{stab}}^{\text{SGD}}(n(k-1) + t - 1; 2k),$$

*where* $w_t^k$ *the t'th SGD iterate of the k'th epoch.*

*Proof.* Fix $t, i \in [n]$, and let $\pi(f_t \leftarrow f(\cdot; i))$ denote the distribution obtained from a random permutation followed by replacing $f_t$ with $f(\cdot; i)$. In addition, denote by $\pi \mid f_t = f(\cdot; i)$ a uniformly

distributed permutation conditioned on $f_t = f(\cdot; i)$. It is easily verified both distributions coincide. Now, by the law of total expectation;

$$
\begin{aligned}
\mathbb{E}_{f_1 \ldots f_n} \left[ f_t(w_t^k) \right] &= \frac{1}{n} \sum_{i=1}^{n} \mathbb{E}_{f_1 \ldots f_n} \left[ f(w_t^k; i) \mid f_t = f(\cdot; i) \right] \\
&= \frac{1}{n} \sum_{i=1}^{n} \mathbb{E}_{f_1 \ldots f_n \sim \pi | f_t = f(\cdot; i)} \left[ f(w_t^k; i) \right] \\
&= \frac{1}{n} \sum_{i=1}^{n} \mathbb{E}_{f_1 \ldots f_n \sim \pi(f_t \leftarrow f(\cdot; i))} \left[ f(w_t^k; i) \right] \\
&= \frac{1}{n} \sum_{i=1}^{n} \mathbb{E}_{f_1 \ldots f_n} \left[ f(w_t^{k,(i)}; i) \right],
\end{aligned}
$$

where $w_t^{k,(i)}$ denotes the SGD iterate obtained for the datapoint sequence after replacing $f_t^j$ with $f(\cdot; i)$ in all epochs $j \leq k$. Note this means each epoch differs from its original version in either 0 or 2 indexes. Now

$$
\begin{aligned}
\mathbb{E}_{f_1 \ldots f_n} \left[ F(w_t^k) - f_t^k(w_t^k) \right] &= \frac{1}{n} \sum_{i=1}^{n} \mathbb{E}_{f_1 \ldots f_n} \left[ f(w_t^k; i) - f(w_t^{k,(i)}; i) \right] \\
&\leq \max_{i \in [n]} G \left\| w_t^k - w_t^{k,(i)} \right\| \\
&\leq \epsilon_{\text{stab}}^{\text{SGD}} (n(k-1) + t - 1; 2k),
\end{aligned}
$$

which completes the proof. $\qquad\square$

We will also make use of a generalization of Lemma 11 given in [7, Lemma 3.1]. The next lemma is a direct implication of it.

**Lemma 9.** *The generalized uniform-argument-stability (see Eq. (41)) rate of SGD with step-size $\eta > 0$ for $G$-Lipschitz convex functions satisfies*

$$
\epsilon_{\text{stab}}^{\text{SGD}}(\tau; J) \leq 2G\eta\sqrt{\tau} + 4\eta G J.
$$

We are now ready to prove the single-shuffle convergence upper bound.

*Proof of Theorem 6 (single-shuffle case).* Similarly to the multi-shuffle case, we have;

$$
\begin{aligned}
F(\widehat{w}) - F(w^\star) &\leq \frac{1}{nK} \sum_{k=1}^{K} \sum_{t=1}^{n} F(w_t^k) - F(w^\star) \\
&= \frac{1}{nK} \sum_{k=1}^{K} \sum_{t=1}^{n} F(w_t^k) - f_t^k(w^\star) \\
&= \frac{1}{nK} \sum_{k=1}^{K} \sum_{t=1}^{n} F(w_t^k) - f_t^k(w_t^k) + \frac{1}{nK} \sum_{k=1}^{K} \sum_{t=1}^{n} f_t^k(w_t^k) - f_t^k(w^\star) \\
&\leq \frac{1}{nK} \sum_{k=1}^{K} \sum_{t=1}^{n} F(w_t^k) - f_t^k(w_t^k) + \frac{D^2}{2\eta nK} + \frac{\eta G^2}{2},
\end{aligned}
$$

with the last inequality following from the standard $nK$ round regret bound for gradient descent [see e.g., 15]. To bound the other term, we now apply Lemma 8 and Lemma 9 to obtain;

$$
\begin{aligned}
\mathbb{E} \left[ F(w_t^k) - f_t^k(w_t^k)) \right] &\leq G \epsilon_{\text{stab}}^{\text{SGD}}(nk + t; 2k) \\
&\leq 2\eta G^2 (\sqrt{n(k-1) + t} + 4k) \\
&\leq 2\eta G^2 (\sqrt{nK} + 4K).
\end{aligned}
$$

Now,

$$\mathbb{E}\left[F(\widehat{w}) - F(w^\star)\right] \leq \frac{1}{nK}\sum_{k=1}^{K}\sum_{t=1}^{n}\mathbb{E}\left[F(w_t^k) - f_t^k(w_t^k)\right] + \frac{D^2}{2\eta nK} + \frac{\eta G^2}{2}$$

$$\leq \frac{1}{nK}\sum_{k=1}^{K}\sum_{t=1}^{n}(2\eta G^2(\sqrt{nK} + 4K)) + \frac{D^2}{2\eta nK} + \frac{\eta G^2}{2}$$

$$\leq 8\eta G^2(\sqrt{nK} + K) + \frac{D^2}{2\eta nK} + \frac{\eta G^2}{2}$$

$$\leq \frac{6GD}{n^{1/4}K^{1/4}} + \frac{4K^{1/4}}{n^{3/4}},$$

where the last inequality follows from a choice of $\eta = D/(2Gn^{3/4}K^{3/4})$. When $n \geq K$, the above implies

$$\mathbb{E}\left[F(\widehat{w}) - F(w^\star)\right] \leq \frac{10GD}{n^{1/4}K^{1/4}},$$

and concludes the proof. $\qquad\square$

# E   Stability Lemmas

In this section, we provide statements and proofs for several known results relating to stability properties of SGD. For convenience, we repeat the definition of UAS Eq. (4) with notation suitable for SGD;

$$\epsilon_{\text{stab}}^{\text{SGD}}(t) := \max_{f_1,\ldots,f_t,f'}\max_{i\in[t]}\left\|w_{t+1} - w_{t+1}^{(i)}\right\|, \tag{42}$$

where $f_1,\ldots,f_t,f'$ are any sequence of convex Lipschitz losses, $w_{t+1}$ the iterate produced by gradient descent from $w_1 \in W$ on $f_1,\ldots,f_t$, and $w_{t+1}^{(i)}$ the iterate produced from $w_1$ on the same sequence after replacing $f_i$ with $f'$.

The next lemma relates the difference between the without-replacement loss distribution and the full batch objective to the uniform stability rate Eq. (42) of the optimization algorithm in question. For a proof see [35] (where it was originally stated for average stability, which is a weaker notion and thus implies the uniform stability case as well).

**Lemma 10.** *Let $\{f(w;t)\}_{t=1}^{n}$ be a set of n, G-Lipschitz losses, and $F(w) = \frac{1}{n}\sum_{t=1}^{n}f(w;t)$. Then, for a uniformly random permutation $\pi\colon [n] \leftrightarrow [n]$, and $w_1$ independent of $\pi$, it holds that*

$$\mathbb{E}_\pi\left[F(w_t) - f(w_t;\pi(t))\right] \leq \frac{(t-1)G}{n}\epsilon_{\text{stab}}^{\text{SGD}}(t-1),$$

*where $\epsilon_{\text{stab}}^{\text{SGD}}$ is the stability rate of SGD defined in Eq. (42), and $w_t$ the output of SGD on $\{f(w;s)\}_{s=1}^{t-1}$.*

Following are two lemmas providing uniform stability upper bounds for SGD.

**Lemma 11.** *The uniform argument stability of SGD with step size $\eta > 0$ on convex G-Lipschitz losses is bounded as;*

$$\epsilon_{\text{stab}}^{\text{SGD}}(t) \leq 2G\eta\sqrt{t}.$$

For proof of the above lemma, see [7].

Next, we have standard lemmas providing stability of ERM and regularized ERM in respectively strongly convex and general convex problems.

**Lemma 12** (Strongly Convex ERM Stability). *Let $f\colon W \times Z \to \mathbb{R}$ be $\lambda$-strongly convex and G-Lipschitz for all $z \in Z$. Then*

$$\left|\mathbb{E}_{S\sim\mathfrak{z}^n}\left[F(w_S^\star) - \widehat{F}(w_S^\star)\right]\right| \leq \frac{G^2}{\lambda n}$$

*Proof.* Let $\widehat{w}_S := w_S^\star$ denote the empirical risk minimizer, and $\widehat{w}_{S^i}$ the ERM for the training set with the $i$'th index swapped with a fresh sample $z_i'$. We have

$$\left| \mathbb{E}_{S \sim \mathcal{Z}^n} \left[ F(\widehat{w}_S) - \widehat{F}(\widehat{w}_S) \right] \right| = \left| \frac{1}{n} \sum_{i=1}^n \mathbb{E} \left[ f(\widehat{w}_S; z_i') - f(\widehat{w}_{S^i}; z_i') \right] \right|$$

$$\leq \frac{G}{n} \sum_{i=1}^n \mathbb{E} \|\widehat{w}_S - \widehat{w}_{S^i}\| \leq \frac{4G^2}{\lambda n},$$

where the first inequality is the generalization equals average stability (see e.g., [32]), and the last inequality follows since $\widehat{w}_S$ and $\widehat{w}_{S^i}$ minimize $(1/\lambda n)$-objectives that differ in a $2G$-Lipschitz term. $\square$

**Lemma 13** (Regularized ERM Stability). *Let $f : W \times Z \to \mathbb{R}$ be $G$-Lipschitz for all $z \in Z$, and denote the regularized empirical risk minimizer by $\widehat{w}_S^\lambda := \arg\min_{w \in W} \left\{ \widehat{F}(w) + \frac{\lambda}{2} \|w\|^2 \right\}$. Then*

$$\left| \mathbb{E}_{S \sim \mathcal{Z}^n} \left[ F(\widehat{w}_S^\lambda) - \widehat{F}(\widehat{w}_S^\lambda) \right] \right| \leq \frac{G^2}{\lambda n}$$

*Proof.* Let $F^\lambda(w) := F(w) + \frac{\lambda}{2} \|w\|^2$ and define the regularized empirical loss $\widehat{F}^\lambda$ accordingly. Then we have a $\lambda$-strongly convex problem and by Lemma 12,

$$\left| \mathbb{E}_{S \sim \mathcal{Z}^n} \left[ F(\widehat{w}_S^\lambda) - \widehat{F}(\widehat{w}_S^\lambda) \right] \right| = \left| \mathbb{E}_{S \sim \mathcal{Z}^n} \left[ F^\lambda(\widehat{w}_S^\lambda) - \widehat{F}^\lambda(\widehat{w}_S^\lambda) \right] \right| \leq \frac{4G^2}{\lambda n}.$$

$\square$

# F    Auxiliary Lemmas

The following provides standard step size dependent lower bounds for convex optimization. See also [1] where similar claims are made in their Lemma 6.2 and implicit in the proof of their Theorem 6.1.

**Lemma 14.** *For any step-size $\eta > 0$, $T \in \mathbb{N}$ and $d := \lceil 16\eta^2 T^2 \rceil$, there exists a convex optimization problem $h : W \to \mathbb{R}$ where $W \subseteq \mathbb{R}^{d+1}$ is of constant diameter such that*

$$h(\widehat{w}) - \min_{w \in W} h(w) \geq \frac{1}{8} \min \left\{ \frac{1}{\eta T} + \eta, 1 \right\},$$

*and $\widehat{w}$ is any suffix average of $T$ gradient descent step iterates.*

*Proof.* We shall concatenate two objectives; the first is single dimensional and will contribute the $\eta$ term, the second is $d$ dimensional and will contribute the $1/\eta T$ term.

**First objective.** Set $f_1(w) := |w - \eta/4|$. Since we initialize at 0, the iterates will "zig-zag" between $0$ and $-\eta$. Clearly, any average of iterates is at best $\eta/4$ away from zero loss.

**Second objective.** Set $f(w) := \max_{i \in [d]} \{w(i)\}$, and note $w^\star = -\frac{1}{\sqrt{d}} \mathbf{1}$ where $\mathbf{1}$ denotes the all ones vector $\in \mathbb{R}^d$. We initialize SGD at $w_1 = 0 \in \mathbb{R}^d$, and follow the gradient steps $\nabla f(w_t) = e_i$ where $i \in [d]$ is one of the coordinates that satisfy $w_t(i) \geq w_t(j) \ \forall j \in [d]$. Hence, for any $t \in [T]$,

$$\|w_{t+1}\|_1 \leq \|w_t\|_1 + \eta \|\nabla f(w_t)\|_1 = \|w_t\|_1 + \eta \leq \cdots \leq \eta t \leq \eta T.$$

By the pigeonhole principle, this implies there must exist some coordinate $i$ such that $w_{T+1}(i) \geq -\eta T/d$. In addition, for any $i$, $\overline{w}_{\tau:T}(i) \geq w_{T+1}(i)$. Therefore, assuming $8\eta^2 T^2 \geq 1$ we conclude;

$$f(\overline{w}_{\tau:T}) - f(w^\star) \geq -\frac{\eta T}{d} + \frac{1}{\sqrt{d}} \geq -\frac{\eta T}{8\eta^2 T^2} + \frac{1}{4\eta T} = \frac{1}{8\eta T}.$$

In the case where $8\eta^2 T^2 < 1$,

$$f(\overline{w}_{\tau:T}) - f(w^\star) \geq -\eta T + \frac{1}{2} \geq \frac{1}{2} - \frac{1}{2\sqrt{2}} \geq \frac{1}{4},$$

and the result follows. $\square$