# OpenReview forum: "Benign Underfitting of Stochastic Gradient Descent"
_NeurIPS.cc/2022/Conference — NeurIPS 2022 Accept_

### Official Review · Reviewer_56De · 2022-07-04

**Rating:** 7
**Confidence:** 4
**Soundness:** 4 excellent
**Presentation:** 3 good
**Contribution:** 3 good

**Summary:**

This paper studies the empirical risk and population risk (together with generalization) for different variants of SGD in the context of stochastic convex optimization (they also extend the analysis to other settings, i.e., convex in expectation, strongly convex). Their main contribution is to construct an example that (one-pass) SGD exhibits both empirical risk and generalization gap of $\Omega(1)$. Moreover, they show that this phenomenon does not exist if we use with-replacement SGD. Last, they derive upper and lower bounds for without-replacement SGD in the multi-epoch regime.

**Questions:**

Please see the weaknesses part.

**Limitations:**

Please see the weaknesses part.

**Strengths And Weaknesses:**

**Strengths:**
* This paper studies the behavior of SGD in SCO, providing new intuition of the analysis of test error (population risk). By their construction, even in the convex optimization, there exists an instance that provably minimizes population risk, while the empirical risk is of constant level, resulting in a constant generalization gap. This observation questions the rational optimization-generalization decomposition framework (or ERM framework). The main reason, is that SGD does not minimize the empirical loss, which differs a lot in the smooth setting. In a word, this observation might lead to a new type of analysis of statistical learning, which can have a broad impact on the community.
* The other observations, together with their proof techniques are also interesting. For example, the comparison between with and without replacement might provide some intuition on the advantage of multi-pass SGD in terms of its implicit regularization.
* The paper is well-structured and provides good intuition for the proof technique.
**Weakness:**
* The construction of the failure seems artificial and might not be general enough. They rely on a special structure that (to my best knowledge) no realistic learning problem has a similar structure. It would be great if the author can provide some connection between their constructed example and some realistic examples.
* The analysis relies on the averaged solution instead of the last iteration solution, which prevents the general impact of this work. It would be great if the author can at least comment on or conjecture the result for the last iteration case.

---

> ### Author Response · Authors · 2022-08-02
> **Response**
>
>
> Thank you for your thoughtful review and helpful comments. We are happy to hear you appreciate our contribution.
>
> > “this observation might lead to a new type of analysis of statistical learning, which can have a broad impact on the community.”
>
> > "The other observations, together with their proof techniques are also interesting."
>
> > "The paper is well-structured and provides good intuition for the proof technique."
>
> Thank you for the kind words!
>
> > "The construction of the failure seems artificial and might not be general enough. They rely on a special structure that (to my best knowledge) no realistic learning problem has a similar structure."
>
> The context in which we prove our lower bound is general (Lipschitz) convex optimization, where the classical SGD analyses and upper bounds apply.  Our construction **is** general enough to demonstrate that these classical test-convergence results do not follow (in a general way) from the traditional empirical minimization / generalization gap viewpoint, which was our main point.
>
> That said, it is indeed an interesting direction for future work to study the behavior of SGD and its generalization gap under more stringent assumptions on the loss function.
>
> > "The analysis relies on the averaged solution instead of the last iteration solution"
>
> Please note our analyses apply to any suffix average (e.g., Theorem 1, line 203; Theorem 2, line 265). This includes the last iterate, which is just a suffix average of one iterate.

---

> > ### Comment · Reviewer_56De · 2022-08-07
> > **Thanks for your response**
> >
> > Thanks for your response which addressed nearly all of my concerns. Thanks for pointing out that Theorem 1, 2 covers the last iteration solution!

---

> > > ### Author Response · Authors · 2022-08-08
> > > **Thank you**
> > >
> > > Thank you for this message.  We are pleased to have been able to address most of your concerns - we would be glad to try and clarify any remaining ones.  Thanks!

---

### Official Review · Reviewer_6Rm5 · 2022-07-11

**Rating:** 7
**Confidence:** 2
**Soundness:** 3 good
**Presentation:** 2 fair
**Contribution:** 3 good

**Summary:**

The paper shows, by considering the stochastic convex optimization framework, that there exist problem instances where the SGD solution exhibits both empirical risk and generalization gap of $\Omega(1)$ (in the without replacement case). SGD is therefore not algorithmically stable. This phenomenon does not occur for SGD with replacement.

**Questions:**

- discussions and conclusions are not present

Minors:
- l338: forgo

**Limitations:**

Limitations and potential negative societal impact of this work may be discussed more

**Strengths And Weaknesses:**

Strengths:
- the phenomenon reported in the paper is novel, intriguing and potentially impactful in the context of SGD generalization bounds
- analytical evidence is solid
- the multi-epoch regime is addressed

Weaknesses:
- nor discussions nor conclusions are present

---

> ### Author Response · Authors · 2022-08-02
> **Response**
>
> Thank you for your review.
>
> > "the phenomenon reported in the paper is novel, intriguing and potentially impactful in the context of SGD generalization bounds"
>
> Thank you for appreciating the paper’s contributions!  In light of your supportive comments, we are wondering what was the rationale for your low merit score.  If you have any further concerns that relate to the contents of the paper, such as originality, significance of contribution, presentation, or soundness of claims, we will gladly address them in the discussion phase.
>
> > "nor discussions nor conclusions are present"
>
> Please see our elaborate introduction section for an extensive discussion of the implications of our results and connections to other contemporary research.  We will be very happy to hear if you have more specific comments about particular points in the discussion.

---

### Official Review · Reviewer_6KpT · 2022-07-14

**Rating:** 4
**Confidence:** 3
**Soundness:** 4 excellent
**Presentation:** 2 fair
**Contribution:** 2 fair

**Summary:**

The authors provide a stochastic convex optimisation framework in which one passe SGD (classically) minimises the population risk at a $O(1 / \sqrt{n})$ rate but however exhibits a $\Omega(1)$ training error and generalisation error.

**Questions:**

N/A

**Strengths And Weaknesses:**

The paper is well written. Each result is clear, introduced and commented. The analysis is rigorous and non trivial. Overall I have no criticism concerning the technical results, which I believe are original. However the main results in section 3 seem anedoctal in the sense that (1) nobody does one pass SGD (2) the considered setting is very peculiar and clearly not encountered on a daily basis: cherry picked distribution and loss, $d \geq 2^{n \log n}$.  This on its own would not bother me (as mentioned, I find the results original and interesting), however my main concern are the conclusions / interpretations which are drawn from the results.  Overall I find that the introduction is very misleading, here are several claims in it which I believe are misleading / false:

- *"First, it is clear that SGD cannot be framed as any reasonable regularized empirical risk minimization procedure for the simple reason that it does not minimize the empirical risk, **which challenges the implicit regularization viewpoint to the generalization of SGD**"*: the implicit regularization viewpoint is crucial in the interpolation setting (setting which is met in practice). Hence I cannot agree with your statement: it does not challenge the viewpoint as you consider a totally different setting.

- *"Second, any attempt to explain generalization of SGD by uniform convergence over any (possibly data-dependent) hypotheses set cannot hold, simply because the sample average associated with the very same training set SGD was trained on is not necessarily close to its respective expectation"*:  (1) your result holds only for one pass SGD (which nobody does in practice), hence one could expect the classical train / generalisation gap to hold for multiple passes

- in the abstract: *"Consequently, it turns out that SGD is not algorithmically stable in any sense, and its generalization ability cannot be explained by uniform convergence or any other currently known generalization bound technique for that matter (other than that of its classical analysis)"*: again this only holds for one pass SGD which is never considered in practice.


Minor comments:
- in the population risk convergence, it would have been clearer to put the exact convergence bound instead of a big $O( )$ so that it is clear that there is no hidden dependence in the dimension $d$ which would have killed the rate.
- a conclusion would have been appreciated
- line 183: $w^*_S = \arg \min _w \hat{F}(w)$: why is the argmin unique ? if not unique which one do you choose ?

---

> ### Author Response · Authors · 2022-08-02
> **Response**
>
> Thank you for your review and helpful comments. Below, we hope to have addressed all of your concerns.
>
> > "The paper is well written. Each result is clear, introduced and commented. The analysis is rigorous and non trivial. Overall I have no criticism concerning the technical results, which I believe are original."
>
> Thank you for the kind words.
>
>
> ### Main replies
>
> > Recurring claims regarding one-pass SGD not being used in practice
>
> We focus on one-pass SGD not because it is used in practice, but because it is the prototypical variant most extensively studied in **theory**, ever since Nemirovski & Yudin (1983)---e.g. refs [1,2,3,4,5] below. Our main and very surprising finding (at least to us) is that the generalization of this method, very well known in the community, does not adhere to the very basic principles of learning theory (as we discuss in the introduction).
>
> In addition, while this is perhaps less relevant to our main point here, please note that **we actually do address the multi-pass case** in Theorems 5 and 6, where we prove upper and lower bounds on the empirical risk of multi-epoch SGD (extending these results to a generalization gap lower bound is not of much interest because multi-epoch SGD does not enjoy optimal test loss bounds in the framework we consider).
>
> > "my main concern are the conclusions / interpretations which are drawn from the results. Overall I find that the introduction is very misleading"
>
> Note we mention already in the abstract the precise variant of SGD we consider: one-pass SGD. All claims made in the paper, and in the introduction in particular, were clearly stated in the context of the stochastic, convex one-pass SGD setup (excluding Section 5, Theorems 5 and 6 which pertain to the multi-epoch setup). That said, our study is definitely **inspired** by broader questions and we do discuss these questions in our paper, but as far as our claims and results go, all are within the scope of the convex one-pass setup, and we very much emphasized this.
>
> Any broader interpretations of our results alluded to in your review, to which we reply in more detail below, were not claimed in our paper—and in fact, we tried to be very careful in avoiding such misinterpretations. However, if you still feel that in certain places we were too vague, please let us know in your final review and we will fix accordingly.
>
>
> ### Additional replies
>
> > "First…. the implicit regularization viewpoint is crucial in the interpolation setting (setting which is met in practice). Hence I cannot agree with your statement";
>
> > "Second… your result holds only for one pass SGD (which nobody does in practice), hence one could expect the classical train / generalization gap to hold for multiple passes"
>
> Our statements were restricted to the stochastic, convex, one-pass setup. This is clearly stated one sentence prior to the ones you have quoted, lines 64-65: *"Many previously plausible explanations for generalization properties of this algorithm are thereby rendered inadequate, at least **in the elementary convex setup we consider here**"*
>
> At a higher level, we are not sure what you meant by “crucial in the interpolation setting”. Perhaps you would like to suggest relevant references, in which case we will gladly cite and make the required comparisons and discussions. But again, we are not claiming there is or there isn’t such a thing in any setup other than the one we study in the paper (if anything, we raise it as a question).
>
> > “in the abstract… again this only holds for one pass SGD which is never considered in practice.”
>
> Note that the abstract actually clearly states we consider one-pass SGD !
>
> > “the considered setting is very peculiar and clearly not encountered on a daily basis:  cherry picked distribution and loss"
>
> Indeed, our construction is artificial - but this is the case with **any** lower bound / impossibility construction, in optimization and more generally. The point of any impossibility result is not to be “natural” or “real-world” but rather establish that no upper bound can ever exist under the same (or broader) set of assumptions.
>
> > Minor comments relating to convergence rate, conclusion, and population minimizer
>
> Thanks! we will take these into account for the final version (indeed the population minimizer should be changed to an arbitrary one)
>
> ### References
>
> [1] Moulines, E. and Bach, F. Non-asymptotic analysis of stochastic approximation algorithms for machine learning. Neurips 2011.
>
> [2] Rakhlin, A., Shamir O., and Sridharan K. Making gradient descent optimal for strongly convex stochastic optimization. ICML 2012.
>
> [3] Shamir, O. and Zhang, T. Stochastic gradient descent for non-smooth optimization: Convergence results and optimal averaging schemes. ICML 2013.
>
> [4] Hazan E. Introduction to online convex optimization. Foundations and Trends in Optimization 2016.
>
> [5] Shai Shalev-Shwartz and Shai Ben-David. Understanding machine learning: From theory to algorithms 2014.

---

### Official Review · Reviewer_1m1F · 2022-07-18

**Rating:** 5
**Confidence:** 2
**Soundness:** 3 good
**Presentation:** 3 good
**Contribution:** 2 fair

**Summary:**

The authors study SGD, with and without replacement, and construct counterexamples for which SGD fails to converge despite the objective being otherwise well-behaved (e.g. convex and smooth).

**Questions:**

None

**Strengths And Weaknesses:**

This appears to be a significant work but is inappropriate for the NeurIPS community because it is too theoretical. The dataset Z and the objective function (bottom pg. 5) are simple but it is unclear how this result relates to real-world machine learning problems. This is evidenced by the lack of experiments in the paper.

---

> ### Author Response · Authors · 2022-08-02
> **Response**
>
> Thanks for your review. We understand that your only concern is the paper being too theoretical; below, we hope to have addressed this.
>
>
>
> If you have any concerns that relate to the contents of the paper, such as originality, significance of contribution, presentation, or soundness of claims, we will be more than happy to hear and discuss.
>
>
>
> > "This appears to be a significant work”
>
>
>
> Thank you for noting!
>
>
>
> > "but is inappropriate for the NeurIPS community because it is too theoretical."; "This is evidenced by the lack of experiments in the paper."
>
>
>
> Convex optimization, and in particular stochastic convex optimization are not considered by any Neurips standard too theoretical. Papers similar to ours that obtain lower and upper bounds for generalization/optimization in this framework are being accepted to Neurips and ICML on a regular basis (e.g., references [1, 2, 3, 4] below, to name a very few). In the vast majority of these papers there are no experiments.
>
>
>
> > "The dataset Z and the objective function (bottom pg. 5) are simple but it is unclear how this result relates to real-world machine learning problems.";
>
>
>
> Indeed, our construction is artificial - but this is the case with **any** lower bound / impossibility construction, in optimization and more generally. The point of any impossibility result is not to be “natural” or “real-world” but rather establish that no upper bound can ever exist under the same (or broader) set of assumptions.
>
>
>
>
> References:
>
>
>
> [1] Agarwal, A., Wainwright, M.J., Bartlett, P. and Ravikumar, P. Information-theoretic lower bounds on the oracle complexity of convex optimization. Advances in Neural Information Processing Systems, 22, 2009.
>
>
>
> [2] Rakhlin, A., Shamir O., and Sridharan K. Making gradient descent optimal for strongly convex stochastic optimization. In International Conference on International Conference on Machine Learning. 2012.
>
>
>
> [3] Feldman, V. Generalization of erm in stochastic convex optimization: The dimension strikes back. Advances in Neural Information Processing Systems, 29, 2016.
>
>
>
> [4] Allen-Zhu, Z. How to make the gradients small stochastically: Even faster convex and nonconvex sgd. Advances in Neural Information Processing Systems, 31, 2018.

---

### Meta-Review · Area_Chair_i6K7 · 2022-08-27

**Recommendation:** Accept
**Confidence:** Less certain

**Metareview:**

This paper analyzes the behavior of SGD for stochastic convex optimization, showing that there exist problem instances where the SGD solution exhibits both significant empirical risk and generalization gap in the without replacement case. The finding is potentially impactful in the context of SGD generalization bounds. The paper is well-structured and provides good intuition for the proof technique. However, I encourage the author(s) to provide a construction of the failure that is more general and less artificial.

**Award:**

No

---

### Decision · Program_Chairs · 2022-09-14

Accept